# Protein aggregates are associated with replicative aging without compromising protein quality control

**Juha Saarikangas\*, Yves Barral\***

Institute of Biochemistry, Eidgenössische Technische Hochschule Zürich, Zürich, Switzerland

**Abstract** Differentiation of cellular lineages is facilitated by asymmetric segregation of fate determinants between dividing cells. In budding yeast, various aging factors segregate to the aging (mother)-lineage, with poorly understood consequences. In this study, we show that yeast mother cells form a protein aggregate during early replicative aging that is maintained as a single, asymmetrically inherited deposit over the remaining lifespan. Surprisingly, deposit formation was not associated with stress or general decline in proteostasis. Rather, the deposit-containing cells displayed enhanced degradation of cytosolic proteasome substrates and unimpaired clearance of stress-induced protein aggregates. Deposit formation was dependent on Hsp42, which collected non-random client proteins of the Hsp104/Hsp70-refolding machinery, including the prion Sup35. Importantly, loss of Hsp42 resulted in symmetric inheritance of its constituents and prolonged the lifespan of the mother cell. Together, these data suggest that protein aggregation is an early aging-associated differentiation event in yeast, having a two-faceted role in organismal fitness.

**\*For correspondence:** juha. saarikangas@bc.biol.ethz.ch (JS); yves.barral@bc.biol.ethz.ch (YB)

**Competing interests:** The authors declare that no competing interests exist.

## Introduction

Aging results in an increasing decline of the organism's fitness over time (*Lopez-Otin et al., 2013*). Remarkably, this process segregates asymmetrically during budding yeast division: the mother cell forms an aging lineage, whereas the daughters generated by these mothers rejuvenate to form eternal lineages, similar to the segregation of soma and germ lineages in metazoans. Such lineage separation requires that the inheritance of factors that promote aging, such as defective/deleterious organelles, proteins, and DNA, is asymmetric during cell division (*Sinclair and Guarente, 1997*; *Aguilaniu et al., 2003*; *Erjavec et al., 2007*; *Henderson and Gottschling, 2008*; *Shcheprova et al., 2008*; *Liu et al., 2010*; *Zhou et al., 2011*; *Hughes and Gottschling, 2012*; *Higuchi et al., 2013*; *Clay et al., 2014*; *Denoth Lippuner et al., 2014*; *Henderson et al., 2014*; *Higuchi-Sanabria et al., 2014*; *Thayer et al., 2014*; *Katajisto et al., 2015*). Therefore, how cells are able to recognize, sort, and coordinate the asymmetric segregation of aging factors and other fate determinants is an outstanding question in biology (*Neumuller and Knoblich, 2009*).

Protein aggregates and/or damaged proteins are a hallmark in the etiology of many human disorders associated with aging (*Hartl et al., 2011*; *Wolff et al., 2014*), and their presence correlates with aging of mitotically active yeast and drosophila stem cells (*Aguilaniu et al., 2003*; *Erjavec et al., 2007*; *Bufalino et al., 2013*; *Coelho et al., 2013*). Studies on budding yeast have shown a correlation between the accumulation of protein aggregates and replicative aging by demonstrating that Hsp104-mediated protein disaggregation is required for full replicative life span (*Erjavec et al., 2007*), and that over-expression of Mca1, which counteracts the formation of stress- and age-associated protein aggregates (*Lee et al., 2010*; *Hill et al., 2014*), extends the life span of yeast mother cells (*Hill et al., 2014*).

**eLife digest** Aging is a complex process. Studies involving a single-celled organism called budding yeast are commonly used to investigate the factors that contribute to aging. When these yeast cells divide, a small daughter cell buds out from a large mother cell. A mother cell has a limited lifespan and produces a finite number of daughter cells and then dies (a phenomenon referred to as 'replicative aging'). However, when a daughter cell forms, the daughter's age is reset to zero, giving it the full potential to produce new offspring.

Previous research on budding yeast has shown that damaged or aggregated proteins accumulate in the mother cells but not the daughter cells, and that this accumulation of proteins contributes to shortening the lifespan of the mother cell. Furthermore, protein aggregation has also been associated with a number of age-related diseases in humans, including neurodegenerative disorders such as Alzheimer's and Parkinson's disease. However, it remains unclear how cells respond to protein aggregation that occurs during aging.

Many studies that have previously investigated this question have relied on exposing cells to stressful conditions, such as high temperatures, in order to trigger proteins to aggregate. But now, Saarikangas and Barral have studied how proteins aggregate under normal, unstressed conditions in budding yeast as they age. The experiments revealed that most unstressed yeast cells develop a single deposit of aggregated proteins already during early aging. This age-associated structure proved to be a different type of response than the protein aggregation that occurs during stress.

Furthermore, the deposit did not form as a consequence of the cell having obvious problems with folding its proteins, nor did the deposit hinder cells from coping with stressful conditions that trigger protein misfolding. Rather, this deposit supported the ability of the cell to degrade defective proteins. This suggests that the deposit represents an early adaptive response to aging, which might consequently provide aged cells some advantage over their younger counterparts.

Saarikangas and Barral also found that this protein deposit was always retained in the mother cell and not passed onto the daughters at cell division. Further experiments showed that an enzyme called heat shock protein 42 was responsible for collecting target proteins and bring them together to form the single deposit. Reducing the levels of this enzyme prevented the deposit from forming and extended the lifespan of the mother cells. Thus, these findings suggest that mother cells collect harmful protein aggregates into a single deposit and prevent them from entering the daughter cells. Further work is needed to understand how the deposit is preferentially retained within the mother cell.

How cells respond to protein aggregation that occurs specifically during aging has remained elusive since most studies investigating the cellular responses to protein aggregation have relied on over-expression of non-native, aggregation prone proteins, proteostasis inhibitors, or other stressors, such as heat (*Kaganovich et al., 2008*; *Liu et al., 2010*; *Specht et al., 2011*; *Zhou et al., 2011*; *Malinovska et al., 2012*; *Spokoini et al., 2012*; *Winkler et al., 2012*; *Escusa-Toret et al., 2013*; *Zhou et al., 2014*). These studies have uncovered specific modes of cytosolic compartmentalization that take place when cells encounter proteotoxic stress. For example, cells stressed with heat respond by forming multiple protein aggregates (referred to as peripheral aggregates, stress foci, Q-bodies, or CytoQ) at the surface of the ER (*Specht et al., 2011*; *Spokoini et al., 2012*; *Escusa-Toret et al., 2013*; *Miller et al., 2015*; *Zhou et al., 2014*; *Wallace et al., 2015*). These structures, hereafter referred as Q-bodies, contain acutely misfolded proteins that are sorted between the nuclear and cytoplasmic degradation/deposit sites by the Hook family proteins Btn2 and Cur1 (*Malinovska et al., 2012*), and coalesce together by the aid of small heat shock proteins; Hsp42 in budding yeast (*Specht et al., 2011*; *Escusa-Toret et al., 2013*), and Hsp16 in fission yeast (*Coelho et al., 2014*). Simultaneously, Q-bodies are being rapidly resolved by the protein disaggregase Hsp104 (*Parsell et al., 1994*; *Specht et al., 2011*; *Zhou et al., 2011*; *Spokoini et al., 2012*; *Escusa-Toret et al., 2013*), together with other heat shock responsive chaperones such as Hsp70 and Hsp82 (*Escusa-Toret et al., 2013*). The formation of Q-bodies seems to aid stress survival, as the deletion of *HSP42* resulted in defective tolerance of prolonged heat stress (*Escusa-Toret et al., 2013*). The asymmetric inheritance of Q-bodies by the mother cells is promoted by the geometry of the bud neck (*Zhou et al., 2011*),

tethering to mitochondria (*Zhou et al., 2014*), and by actin cable-mediated retrograde transport, which is dependent of Hsp104 and Sir2 (*Liu et al., 2010*; *Song et al., 2014*). Notably, Sir2 is also a key player in processes that underlie the asymmetric segregation of damaged mitochondria (*Higuchi et al., 2013*) and the accumulation of extrachromosomal DNA circles (*Sinclair and Guarente, 1997*; *Kaeberlein et al., 1999*) to the aging mother cell.

Prolonged Q-body-inducing stress (heat or over-expression of thermolabile proteins) combined with proteasome inhibition can lead to the formation of a dynamically exchanging deposit of ubiquitylated proteins named the juxtanuclear quality compartment, JUNQ (*Kaganovich et al., 2008*; *Escusa-Toret et al., 2013*). This structure is regulated by the Upb3 deubiquitinase (*Oling et al., 2014*), by proteosomal activity (*Andersson et al., 2013*) and by lipid droplets (*Moldavski et al., 2015*), and it was also shown to appear during replicative aging (*Oling et al., 2014*). The faithful inheritance of this structure by the mother cell is dependent on its association with the nucleus (*Spokoini et al., 2012*). More recently, it was shown that the 'JUNQ' might actually reside inside the nucleus, and it was thus renamed as intranuclear quality control compartment, INQ (*Miller et al., 2015*). The JUNQ/INQ assembly is dependent on Btn2-aggregase (*Miller et al., 2015*), a protein also found to be involved in prion curing (*Kryndushkin et al., 2008*, *2012*). Apart from the JUNQ/INQ structure, terminally aggregating proteins, such as the amyloidogenic prions Rnq1 and Ure2, were shown to partition to an non-dynamic, vacuole-associated deposit called the insoluble protein deposit IPOD (*Kaganovich et al., 2008*; *Tyedmers et al., 2010b*), which has remained less well characterized.

Despite this wealth of data, it remains unclear how these exogenous/stress-induced aggregation models relate to protein aggregation that takes place during physiological 'healthy' aging. Particularly, it is unclear why/how protein aggregates arise during aging, how are they segregated during cell division and, importantly, what is their consequence to the protein quality control of the aging cell, as well as to the aging process itself. To illuminate these aspects, we probed the role of protein aggregation during unperturbed replicative aging. Our findings indicate that protein aggregation is a prevalent and highly coordinated event of early aging and is not solely associated with proteostasis deterioration. Instead, we provide evidence that age-associated protein aggregation may initially benefit the cytosolic protein quality control, but eventually becomes involved with age-associated loss of fitness.

## Results

### Formation of a protein deposit during early replicative aging

To address the role of protein aggregation in unperturbed, physiological aging, we analyzed microscopically the replicative age-associated protein aggregation landscape in budding yeast by visualizing different chaperone proteins that mark aberrantly folded and aggregated proteins. By employing the Mother Enrichment Program (MEP) (*Lindstrom and Gottschling, 2009*) (*Figure 1—figure supplement 1A*), we harvested cells of different age and first analyzed the localization of endogenous GFP-tagged protein-disaggregase Hsp104 (*Parsell et al., 1994*; *Glover and Lindquist, 1998*), a broad sensor for protein aggregates (*Figure 1A*, *Haslberger et al., 2010*). Interestingly, we found many cells displaying an aggregate (typically a single bright Hsp104-labeled focus) and this portion increased in a progressive, age-dependent manner such that >80% of cells that had undergone more than 6 divisions displayed such a structure (*Figure 1A,B*), as previously reported (*Aguilaniu et al., 2003*; *Erjavec et al., 2007*). Co-localization analysis with Hsp104 demonstrated that the Hsp70 proteins Ssa1 and Ssa2, the small heat shock protein Hsp42, and the Hsp40 protein Ydj1 readily localized to these aggregates, the Hsp26 was found to be enriched in only 15% of Hsp104-labeled foci, while no accumulation of Hsp40 protein Sis1 or the Hsp90 protein Hsp82 was detected (*Figure 1C*, *Figure 1—figure supplement 2A–C*). Importantly, age-dependent appearance of these structures was also detected in diploid cells, in other strain backgrounds (W303), independently of the MEP procedure, and when different fluorophores where used for tagging Hsp104 (*Figure 1—figure supplement 2D–H*), indicating that their formation represents a general, age-dependent phenomenon of budding yeast cells.

To characterize the nature of these protein deposits further, we performed live-cell imaging by acquiring images every 15 min over several hours and covering the entire depth of the cell. This showed that most mother cells started to form this aggregate already after budding three to four times (*Figure 1D*, 390 min). The fastest growth phase of the aggregate took place during the first half

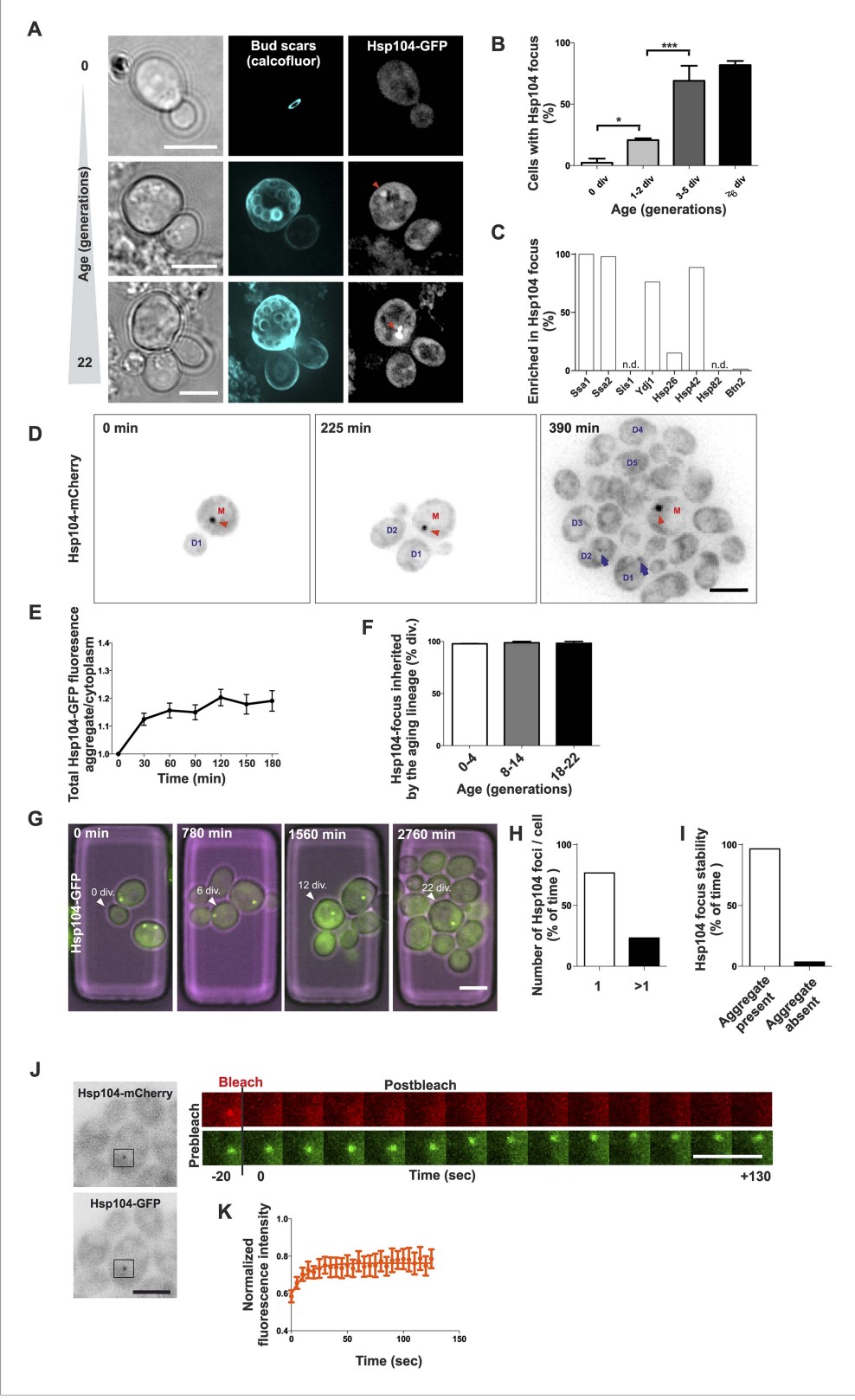

**Figure 1**. Replicative aging leads to the formation of age-associated protein deposit. (**A**) Representative images of cells expressing endogenous Hsp104 tagged with GFP. The C-terminal tagging does not hamper Hsp104 disaggregation activity (*Specht et al., 2011*). Cells of different age were harvested using the Mother Enrichment

*Figure 1. continued on next page*

*Figure 1. Continued*

Program (MEP) (*Lindstrom and Gottschling, 2009*) and stained with calcofluor. (**B**) Percentage of cells of different age groups containing at least one Hsp104-focus, (N = 135 to 472 cells per age group). (**C**) Fraction of Hsp104-mCherry foci that are enriched with the indicated chaperones (N = 16 to 92 Hsp104-focus containing cells per strain, n.d. = not detected). (**D**) Representative frames of a movie of dividing cells expressing Hsp104-mCherry (black). Red arrowhead indicates an aggregate that is retained in the mother cell (M). This cell divided five times giving rise to four daughter cells that start to form aggregates after dividing 2–3 times (blue arrows at 390 min). (**E**) Integrated density at newly forming age-associated protein deposits (deposit/cytoplasm) over time, (N = 26). (**F**) Fraction of divisions during which the age-associated protein deposit is asymmetrically inherited by the mother cell as a function of the age of the mother cell (N = 66 to 306 divisions per age group from 2-3 independent experiments. The approximate mother cell age was estimated from separate bud scar analysis). (**G**) Representative micrographs of a dividing mother cell expressing Hsp104-GFP followed for 66 hr in a microfluidic chip (*Lee et al., 2012*).
(**H**) Proportion of cells having the indicated number of Hsp104 foci. Hsp104 foci containing cells that could be followed >10 consecutive divisions were quantified for the number of Hsp104-foci/cell at each time point, (N = 44). (**I**) Total time spent with and without an Hsp104-focus for cells undergoing >10 consecutive divisions, starting with a focus, (N = 44). (**J**) Fluorescent recovery after photobleaching (FRAP) analysis of Hsp104-mCherry turnover at age-associated protein deposit. The mCherry signal at the age-associated protein deposit was photobleached in *HSP104-GFP/HSP104-mCherry* diploid cells and the kinetics of recovery were monitored, using the GFP signal to localize the age-associated protein deposit over time. (**K**) Fitting of nine recovery curves showed that on average a large (59%) fraction was immobile and the half-time recovery for the mobile fraction was 8.9 s, (N = 9). Scale bars 5 μm. Graphs display mean ± SEM, *p < 0.05, ***p < 0.001.

The following figure supplements are available for figure 1:

**Figure supplement 1**. Schematic representation of the strategies used here to study aged cells.

**Figure supplement 2**. Age-associated protein deposit formation is a general age-dependent phenomenon marked by a subset of chaperones.

an hour following its detection, after which the Hsp104 signal intensity increased only marginally (*Figure 1D,E*). While the aggregates initially underwent occasional dissolution, they became stable within a few hours from nucleation (*Figure 1E*). Pedigree analysis of aggregate inheritance in cells of different age showed that they faithfully segregated to the aging mother cell (98% of divisions), irrespective of its age (*Figure 1D,F*, *Video 1*). To further analyze the persistence and behavior of the deposits over the entire replicative life span of cells, we used a microscope-coupled microfluidic dissection platform (*Lee et al., 2012*) (*Figure 1—figure supplement 1B*). This showed that the aggregate was efficiently maintained as one compartment, and whenever new foci emerged, they typically merged soon after with the pre-existing deposit. We quantified the number of Hsp104 foci (1 or >1) and its post-nucleation stability in cells that were tracked for at least 10 divisions after the aggregate had appeared. These cells preferentially (>80% of time) displayed only a single Hsp104 focus (*Figure 1G–H*), which was very stable, typically persisting until the last divisions of the cell (*Figure 1G,I*). Moreover, this structure was largely non-dynamic, displaying limited exchange of Hsp104 with the cytoplasm, as determined by fluorescent recovery after photobleaching (FRAP) analysis in diploid cells that expressed one GFP- and one mCherry-tagged copy of *HSP104* (*Figure 1J*). We bleached and measured the recovery of the mCherry signal (*Figure 1J*) and by fitting nine recovery curves, found that on average a large fraction (59%) of Hsp104 was immobile, while the mobile fraction displayed a half-time recovery rate of 8.9 s (*Figure 1K*). Together, these data show that a large majority of unstressed yeast cells develop a protein deposit early during replicative aging. This deposit displays limited exchange with the cytoplasm (assessed by Hsp104), is efficiently maintained as a single compartment, and is faithfully inherited by the aging mother cell.

## The physiological constituents of the age-associated protein deposit include prion protein Sup35

To better understand how the age-associated protein deposit forms and what are its physiological constituents, we selected two proteins containing glutamine- and/or asparagine-rich domains, Sup35

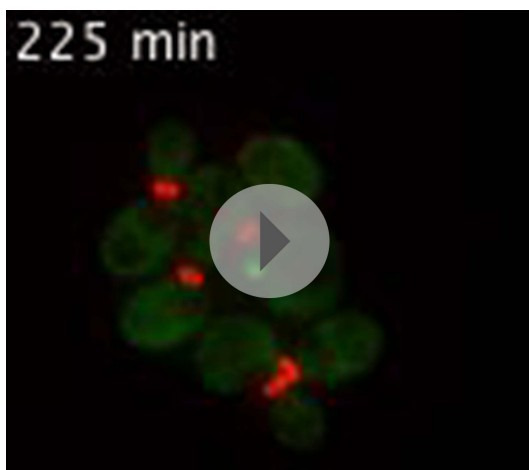

**Video 1.** Age-associated protein deposit is a stable structure that is faithfully inherited by the aging mother cell lineage. Aged yeast cell expressing Hsp104-mCherry (in green) to mark protein aggregates and Cdc10-GFP (in red) to mark the mother bud interface undergoing four consecutive divisions.

and Dcp2, which are known to undergo conformational switches and to aggregate, and we monitored their localization in respect to the age-associated protein deposit. Sup35 (eRF3) is a translation termination factor that can undergo stable amyloid-like prion conversion from non-prion [*psi*–] to prion state [*PSI*+] (*Chernoff et al., 1993*; *Ter-Avanesyan et al., 1994*; *Wickner, 1994*; *Patino et al., 1996*; *Paushkin et al., 1996*). Thus, we tested whether Sup35-GFP is targeted to the age-dependent aggregate and if the prion status plays a role in the targeting to and/or in the formation of the age-associated protein deposit. Importantly, Sup35-GFP clearly accumulated into Hsp104-mCherry-labeled age-associated protein deposit in 33% of [*PSI*+] cells that contained such a deposit, whereas no accumulation was detected in the non-prion [*psi*–] cells (*Figure 2A–C*). As expected, the formation of this bright Sup35 focus was age-dependent (*Figure 2D*), in line with earlier observations (*Derdowski et al., 2010*), but interestingly the [*PSI*+] state did not have an overall effect on Hsp104-labeled deposit forma-

tion (*Figure 2E*). Similarly, the elimination of the [*RNQ*+] prion by deleting *RNQ1* did not negatively influence the formation of the age-associated deposit (data not shown). Notably, time-lapse imaging showed that Sup35 was recruited into the pre-existing age-associated protein deposit (marked by Hsp104-mCherry). These Sup35-enriched age-associated protein deposits segregated asymmetrically towards the aging mother cell in 98% of mitoses (*Figure 2F,G*). These data provide evidence that the Hsp104-labeled focus is a *bona fide* deposit site for aggregating proteins, and that prion conversion of Sup35 promotes its gradual storage into the age-associated protein deposit, but does not potentiate its formation per se.

In contrast, Dcp2, a Q/N-rich component of the reversible P-body mRNP aggregates (*Reijns et al., 2008*), did not accumulate into the age-associated deposit (*Figure 2H,I*), even after induction of P-body formation by reducing the glucose level in the medium to 0.1% (*Decker et al., 2007*) (0/57 Hsp104 foci with Dcp2-GFP, *Figure 2J*). Furthermore, we never observed age-associated protein deposits and P-bodies fusing with each other (*Figure 2J*). This suggests that P-bodies and age-associated protein deposits have different physicochemical properties, representing two distinct modes of cytosolic sub-compartmentalization (*Hyman et al., 2014*; *Kroschwald et al., 2015*). Altogether, these data show that the substrates of the age-associated deposit are non-random and possibly amyloid-like, which might explain their irreversible nature.

## The age-associated deposit is distinguishable from the previously described protein quality control deposits

During recent years, numerous distinct protein deposits, including Q-bodies, JUNQ/INQ, and the IPOD, have been discovered and thus we wanted to explore whether the age-associated protein deposit matches the identity of any of these structures. We first looked at the behavior of Q-bodies, which assemble in response to heat-stress. In stark contrast to the age-associated aggregates (*Figure 1G,I*), Q-bodies induced upon heat stress (42˚C, 30 min) were transient and readily solubilized after the removal of the stress factor (*Figure 3A*; *Liu et al., 2010*; *Escusa-Toret et al., 2013*; *Wallace et al., 2015*). Comparative analysis of the localization of different chaperone proteins between Q-bodies and the age-associated protein deposit showed that all markers of the age-associated protein deposit were also found to accumulate in Q-bodies (*Figure 3B*). Conversely, we found chaperones such as Hsp82 and Btn2 localizing to Q-bodies (*Figure 3B*) albeit being excluded from the age-dependent protein deposit (*Figure 1C*). Together, the reversible nature and the difference in associated chaperones demonstrate that the Q-bodies and age-associated protein deposits can be distinguished.

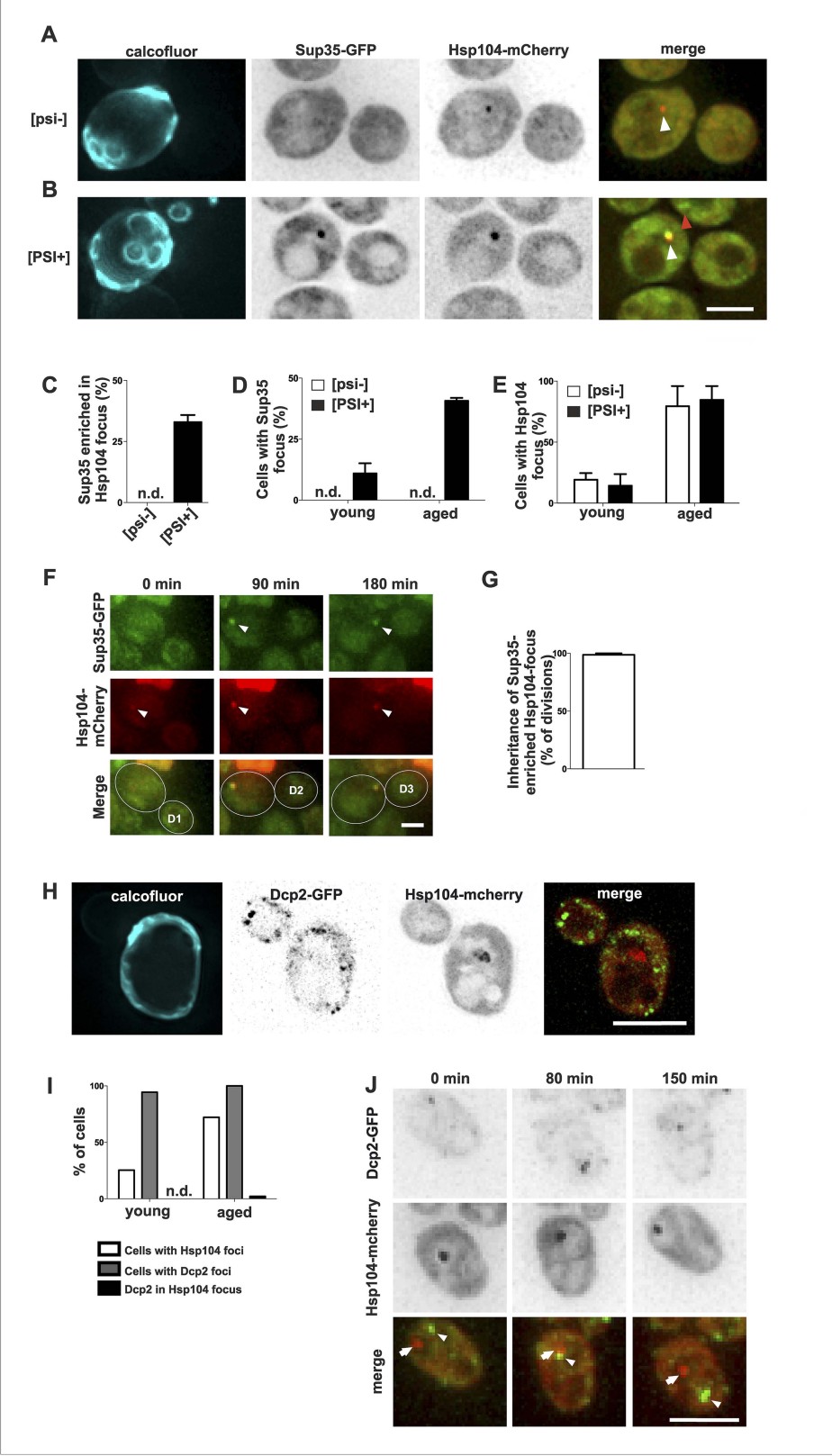

**Figure 2**. Prion form of Sup35 is deposited to the age-associated protein deposit. (**A**, **B**) Co-localization of Sup35-GFP and Hsp104-mCherry in diploid *[psi–]* (**A**) and *[PSI+]* (**B**) cells, where a single locus of the respective gene was tagged with the fluorescent marker indicated. White arrowhead indicates the age-associated protein deposit
*Figure 2. continued on next page*

Figure 2. Continued

(Hsp104 positive) and the red arrowhead Sup35-aggregates not associated with the age-associated protein deposit.
(**C**) Percentage of age-associated protein deposits in [*psi*–] and [*PSI+*] cells with enriched Sup35, (N = 130).
(**D**) Percentage of cells with large Sup35-foci in cells of indicated age groups (N = 157–184 cells pre age group, average age young [*psi*–] 1.1, young [*PSI+*] 1.1, aged [*psi*–] 6.9, aged [*PSI+*] 5.9 generations). (**E**) Percentage of [*psi*–] and [*PSI+*] cells of the indicated age group (see D) containing an age-associated protein deposit, (N = 157–184 cells pre age group). (**F**) Time-lapse images of a [*PSI+*] cell co-expressing Sup35-GFP (green) Hsp104-mCherry (red) at the indicated time points. Arrowheads point at the age-associated protein deposit as observed in the different channels. The newborn daughters are indicated in the bottom row. (**G**) Percentage of divisions where the Sup35-GFP-labeled age-associated protein deposit is retained in the aging mother cell lineage, (N = 204 divisions). (**H**) Fluorescent images of a cell co-expressing the P-body protein Dcp2 tagged with GFP and Hsp104 tagged with mCherry. (**I**) Percentages of cells of indicated age groups that contain the indicated fluorescent foci, (N = 18–71 per group). (**J**) Time-lapse, fluorescent images of Dcp2-GFP and Hsp104-mCherry expressing cells at the indicated time points after switching the cells to 0.1% glucose. Scale bars (**A**, **B**, **H**, **J**) 5 µm, (**F**) 2 µm. Graphs display mean ± SEM.

We then examined whether the age-associated protein deposit would bear the characteristics of the JUNQ/INQ compartment, which is formed during prolonged stress and/or in response to proteasome inhibition. To this end, we induced the GAL-promoter-driven expression of the JUNQ/INQ marker, the unstable proteasome substrate human von Hippel-Lindau tumor suppressor protein (VHL) (*Kaganovich et al., 2008*; *Miller et al., 2015*), and monitored the accumulation of Hsp104-mCherry into the newly forming VHL-foci. VHL typically formed a single focus, which in 87% of the cases did not accumulate Hsp104-mCherry within 30 min after its appearance (*Figure 3C,D*, N = 29). However, the VHL expression was often associated with overall increased Hsp104-mCherry expression over time, suggesting that its expression activates a stress response. Hence, we visualized the JUNQ/INQ without over-expression of exogenous substrates. To accomplish this, we used an Hsp82-GFP, Hsp104-mCherry expressing strain deleted of the *RPN4* gene, leading to increased burden of proteasomal substrates, due to decreased amount of functional proteasomes (*Xie and Varshavsky, 2001*). Indeed, many of *rpn4Δ* cells displayed numerous Hsp82-Hsp104 double-positive Q-body-like puncta (data not shown). In addition, we found cells that displayed two puncta: a Hsp104-Hsp42 double-positive JUNQ-like deposit, and a Hsp104-positive, Hsp82-negative deposit that fills the criteria of the age-associated deposit (*Figure 3E*). To consolidate this further, we investigated the co-localization of the endogenous JUNQ/INQ marker Btn2 (*Miller et al., 2015*) together with the age-associated protein deposit (Hsp104). This analysis showed that in 98.6% of the cases, Btn2 did not localize to the age-associated Hsp104-foci (*Figure 1C* and *Figure 3F*). Moreover, from all thirty identified cells displaying endogenous Btn2 foci, we found only one case in which Hsp104 was enriched at this site (*Figure 3G*, N = 1268 cells). Together, these data suggest that the age-associated protein deposit and the JUNQ/INQ are discrete structures that may exist in parallel.

Finally, we monitored the resemblance between the age-associated protein deposit and IPOD. To visualize cells during IPOD appearance, we imaged Hsp104-mCherry-expressing cells together with the canonical IPOD marker, the galactose-inducible Rnq1-GFP (*Kaganovich et al., 2008*), immediately after placing cells to galactose-containing media. This showed that the IPOD typically appeared as a single focus to which Hsp104 rapidly accumulated (*Figure 3H,I*) (>98% of Rnq1-foci displayed accumulation of Hsp104 within 30 min after their appearance, N = 53). However, by dissecting the Rnq1-GFP appearance dynamics in cell with a pre-existing age-associated protein deposit (marked by Hsp104-mCherry), we found that the aggregating Rnq1-GFP did not accumulate to the age-associated deposit (see white arrowhead in *Figure 3H*), but rather it formed a new aggregate to which Hsp104 then strongly accumulated, while its intensity at the age-associated protein deposit declined (*Figure 3H*). To rule out possible artifacts induced by Rnq1-GFP over-expression, we also looked at the co-localization between the age-associated protein deposit and GFP-tagged endogenous Rnq1 in its prion [*RNQ+*]-state. We analyzed altogether 117 Hsp104 foci-containing [*PIN+, PSI+*] cells and found that in 98.3% of the cases, Rnq1-GFP did not accumulate to this deposit (*Figure 3J*). Within this cohort, we found two cells with an Rnq1-GFP-aggregate, which was in both cases enriched with Hsp104-mCherry (*Figure 3K*). However, both of the Rnq1-aggregate containing cells also displayed an additional Hsp104 focus to which Rnq1 had not accumulated (*Figure 3K*). Together with the results from the Rnq1 over-expression, these data suggest that Rnq1

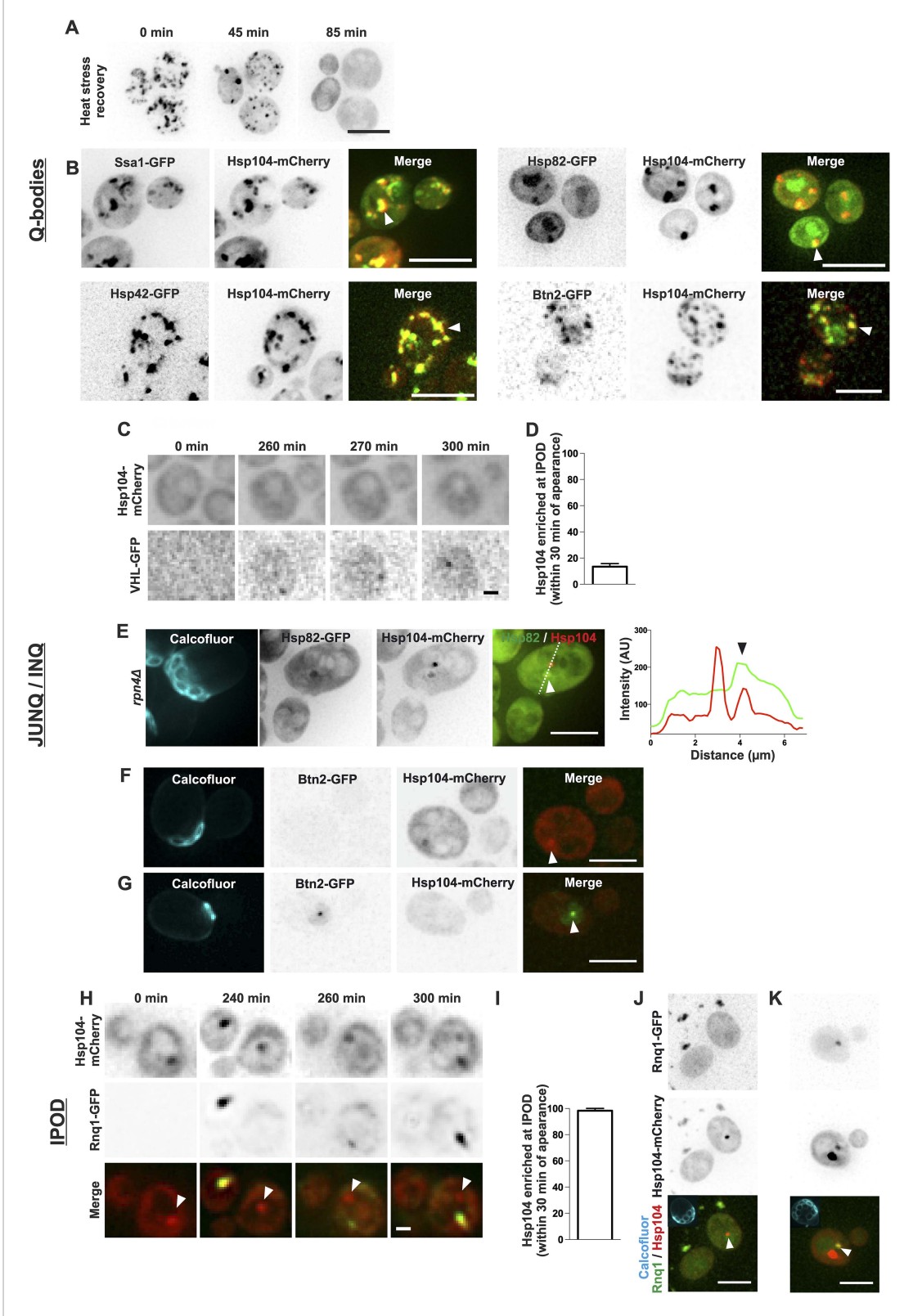

**Figure 3.** The age-associated protein deposit can be distinguished from Q-bodies, JUNQ/INQ, and IPOD. (**A**) Cells were heat shocked (42°C for 30 min) to induce the formation of Q-bodies (stress induced Hsp104-labeled aggregates) and the kinetics of Q-bodies dissolution was followed with time-lapse microscopy of Hsp104-GFP. (**B**) Cells were imaged after heat shock (42°C for 30 min). Arrowhead in 3D-projected images depicts co-localization of Hsp104
*Figure 3. continued on next page*

*Figure 3. Continued*

with age-associated deposit-resident (Ssa1, Hsp42) and non-resident (Hsp82, Btn2) markers in Q-bodies (compare with *Figure 1C*). (**C**) The expression of JUNQ/INQ marker VHL-GFP was induced at the start of the imaging to follow its recognition by Hsp104-mCherry. (**D**) Quantification of VHL-GFP foci recognized by Hsp104-GFP within 30 min of their appearance (N = 29 cells). (**E**) 3D-projected images of *RPN4* deleted cells expressing Hsp82-GFP and Hsp104-mCherry. Hsp82-GFP, which localizes to Q-bodies (*Figure 3A*) but not to age-associated protein deposits (*Figure 1C*), co-localizes with one of the two Hsp104-mCherry foci in aged cell (see fluorescence intensity line-scan over the two foci). (**F**) 3D-projected image of Btn2-GFP and Hsp104-mCherry expressing cell. Btn2 is typically very low abundant and does not accumulate (98.6% of cases, N = 72) to the age-associated deposit (white arrowhead). (**G**) Cell with a Btn2 focus, which typically (96.4% of cases, N = 30) did not display accumulation of Hsp104-mCherry. (**H**) Rnq1-overexpression was induced at the onset of imaging. The panel shows the z-sections displaying the age-associated deposit (white arrowhead in 'merge'). (**I**) Quantification of newly formed Rnq1-GFP foci that accumulate Hsp104-mCherry within 30 min of their appearance (N = 53 cells). (**J**) Representative image of [*PIN+, PSI+*] cell harboring GFP-tagged endogenous Rnq1 and mCherry-tagged Hsp104. Arrowhead indicates the age-associated protein deposit. (**K**) Example of a [*PIN+, PSI+*] cell that displays an Rnq1-aggregate. Arrowhead indicates Hsp104-labeled foci, of which only one has accumulated Rnq1. Scale bars: A-B, E, J-K 5 µm, C, G 2 µm. Graphs display mean ± SEM.

aggregate is likely to represent a structure that is different from the majority of the age-associated protein deposits.

In sum, these data point out marked differences between the age-dependent protein deposit and the previously characterized Q-bodies, IPOD, and the JUNQ/INQ that could derive from context dependent (stress vs aging) differences in cellular responses to protein aggregation.

## Age-associated protein deposits do not compromise the clearance of stress-induced protein aggregates

Appearance of protein aggregates has commonly been associated with defects in protein quality control (*Tyedmers et al., 2010a*). Therefore, we wanted to elucidate if the formation of the age-associated protein deposit is a sign of defective protein quality control. First, we examined whether aged cells display a decline in dealing with proteotoxic stress. As shown before, young cells clear heat-induced protein aggregates (Q-bodies) rapidly after stress removal (*Figure 3A*). Hence, we first wanted to test if this recovery period is affected by the age of the cell (*Figure 4A*). We measured the clearance time of Q-bodies (defined as two or more Hsp104 foci) between young (average age 0.4 generations, between 0 and 1 generations, N = 70) and aged cells (average age 9.2 generations, between 6 and 19 generations, N = 50) following acute heat stress (*Figure 4B*). Surprisingly, no difference in the mean time of protein aggregate clearance was detected between these populations: young: 71 ±5 min old: 72 ±4 min, n.s.) (*Figure 4C*), suggesting that aged cells, despite having a protein deposit, are fit to cope with proteotoxic stress.

We then used a temperature-controlled microfluidic device that enabled us to categorize cells depending on their pre-stress age-associated protein deposit-status, and monitor them under the microscope prior to, during and after undergoing acute proteotoxic stress conditions (*Figure 4D*). The duration of the Q-body response (state in which cells display two or more Hsp104-foci) was plotted over time and demonstrated that cells with and without a pre-existing age-associated deposit showed a comparable Q-body response when heat was applied, while cells that were not exposed to heat did not show similar Q-body response during this time period (*Figure 4E*, N = 53–97 cells). Interestingly, the deposit-containing cells responded slightly faster and to a lesser extend when compared to their clonal counterparts without a pre-existing protein deposit (*Figure 4E*).

We then asked if exposure to proteotoxic stress (Q-body state) would promote the formation of age-associated protein deposits. To this end, we compared the aggregate status of single cells (Hsp104-mCherry: no/one/several puncta) at time points 0 and 380 min, between stress-experienced (between 50 and 80 min) and non-stressed (constantly at 30 °C) cells (*Figure 4F*, N = 32–77 per group). This analysis showed that the majority of non-stressed cells (62.5%) without a deposit at the beginning of the experiment displayed a single aggregate at the end of the experiment (*Figure 4G*), in accordance with the appearance kinetics of age-associated deposit (*Figure 1B,D*). Surprisingly, when the cells without a prior aggregate encountered heat stress (i.e. conditions that induce proteotoxic stress), a substantially smaller portion of them (36.4%) displayed an aggregate at the end of the experiment (*Figure 4G*). Similarly, there was a slight increase of cells without an aggregate in the stress-encountered cohort (17.7% vs 6.8%) among cells that displayed an age-associated deposit

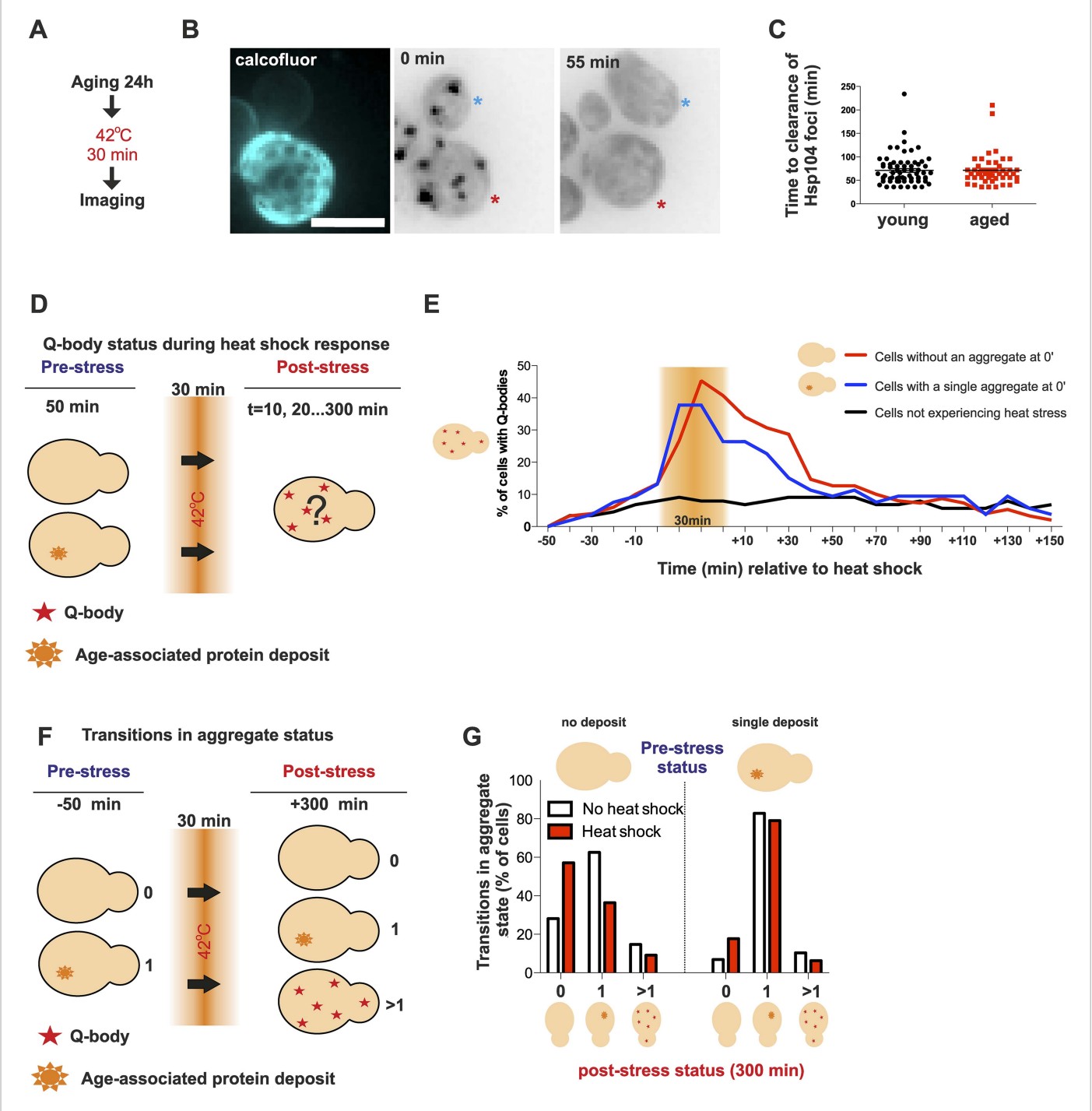

**Figure 4**. Aged cells are not impaired in handling proteotoxic stress. (**A**) Experimental scheme. (**B**) Representative image of displaying the dynamics of recovery from heat shock induced proteotoxic stress (Q-bodies) between aged (red star) and young cell (blue star). (**C**) Q-body clearance time of individual cells of the indicated age groups, (old: average 9.2 generations, age between 6 and 19, N = 50; young: 0.4 generations, age between 0 and 1, N = 70). (**D**) Experimental scheme: Hsp104-mCherry-expressing cells were captured on a temperature-controlled microfluidic device and imaged prior (−50 min), during (30 min) and after (up to 300 min) mild heat shock. It is important to note, that the strength of the heat stress induced on the microfluidic platform is not comparable with *Figure 3A,B* and *Figure 4A–C*. (**E**) Single cell analysis of Q-body formation (cells with >1 Hsp104-foci) in cells with a pre-existing age-associated protein deposit (red line, N = 53), cells without a pre-existing deposit (blue line, N = 97), and cells that did not experience stress (N = 82). Data from two independent experiments. (**F**) Transitions in the aggregate state were recorded 50 min before the heat stress and 300 min after the heat stress. (**G**) Quantification of transitions (N = 32–77 per group, from two independent experiments). White bars indicate cells that did not experience heat stress and red bars denote transitions in aggregate state in heat-stress experienced cells. Scale bar 5 μm.

at the beginning of the experiment (*Figure 4G*). However, irrespectively of stress, cells with an age-associated deposit at the beginning of the experiment preferentially displayed a single deposit at the end of the experiment (non-stressed 82%, stress-experienced 79%) (*Figure 4G*). These results indicate that mild exposure to proteotoxic stress conditions counteract the formation of age-associated protein deposits and demonstrate that cells with an age-associated deposit prior to encountering the stress predominantly reverted back to this state. Altogether, these data indicate that despite forming a protein deposit, aged cells are still fit to coping with acute proteotoxic stress and suggest that the formation of the age-dependent protein deposit is not due to the overload of the general quality control machinery.

## Age-associated protein deposit-containing cells display enhanced degradation of cytosolic ubiquitin–proteasome system substrate

The notion that aged cells with a protein deposit handled proteotoxic stress comparably to their young counterparts prompted us to test the effect of the age-associated deposit on the function of the ubiquitin–proteasome system (UPS), which has an important role in yeast replicative aging (*Kruegel et al., 2011*). Pathological polyQ protein fragments have been shown to impair the UPS function both in yeast and mammalian cells (*Bennett et al., 2005*; *Park et al., 2013*), and UPS substrates have been shown to accumulate in aged cells (*Andersson et al., 2013*). On the other hand, stress-induced protein aggregates (Q-bodies) do not result from failed protein degradation, but may actually benefit the proteostasis of cells undergoing stress (*Escusa-Toret et al., 2013*). We therefore wanted to analyze the contribution of the age-associated deposit on the functionality of the UPS. In order to selectively measure the effect of the age-associated protein deposit on UPS function, we made use of the auxin-inducible degron (AID) system (*Nishimura et al., 2009*), which enabled us to measure the degradation rate of cytosolic (AID-GFP) and nuclear (AID-GFP-NLS) UPS substrates in vivo (*Figure 5A,B*). Cells co-expressing Hsp104-mCherry and AID-GFP were switched to auxin-containing media at the onset of imaging, and the GFP intensities were measured over time from neighboring cells containing or not an Hsp104-puncta. The normalized values were pooled and fitted with a non-linear one-phase decay function. Importantly, this showed that the decay rate of AID-GFP was significantly accelerated in cells with an age-associated deposit when compared to neighboring cells that did not contain an Hsp104-focus (*Figure 5C,D*) (rate constant (K): $0.155 \pm 0.007$ (with) vs $0.129 \pm 0.006$ (without), $p < 0.01$, N = 91 cells/category). However, no difference was detected in the decay rate of nuclear GFP between cells with or without an age-associated protein deposit (*Figure 5E,F*) (K: $0.075 \pm 0.004$ (with) vs $0.079 \pm 0.005$ (without), n.s.), demonstrating that the effect of the age-associated protein deposit on the UPS function is specific to the cytosolic compartment. Collectively, these results suggest that the age-associated deposit promotes cytosolic quality control by enabling more efficient clearance of degradation substrates.

## The age-associated protein deposit is regulated by interplay between Hsp42 and Hsp70/Hsp104 chaperones

Next, we wanted to understand what are the assembly principles of the age-associated deposit. Molecular chaperones generally recognize, refold, or sort aberrantly folded proteins, making them prime candidates to regulate the age-associated protein deposit assembly. Thus, we addressed the contribution of chaperones localizing to the age-associated protein deposit to its formation. Time-lapse microscopy showed that in 24 out of 25 cells, Hsp42 was at the age-associated protein deposit before Hsp104 and was the first protein we find to mark this structure (*Figure 6A*). Ssa1 and Hsp104 were recruited subsequently, appearing concurrently up to 120 min after appearance of the Hsp42-focus (*Figure 6A* and data not shown). To identify the role of these chaperones in the assembly of the age-associated protein deposit, we investigated the consequences of their deletion. Interestingly, loss of the Hsp70 proteins Ssa1 and Ssa2, which act synergistically with the disaggregase Hsp104 in protein refolding (*Glover and Lindquist, 1998*; *Kim et al., 2013*), resulted in rapid formation of the Hsp104 focus (*Figure 6B,C*) and a similar, but milder effect was detected upon deletion of *HSP26*. In stark contrast, deletion of *HSP42* abolished age-associated protein deposit formation (*Figure 6B,C*), which parallels to its function in Q-body assembly (*Specht et al., 2011*; *Escusa-Toret et al., 2013*). To probe the role of Hsp104 and Hsp42 further, we analyzed the consequences of their over-expression on age-associated protein deposit formation (monitored by endogenous Hsp104-GFP). Over-expression of Hsp42 caused precocious nucleation of age-associated protein deposit and promoted

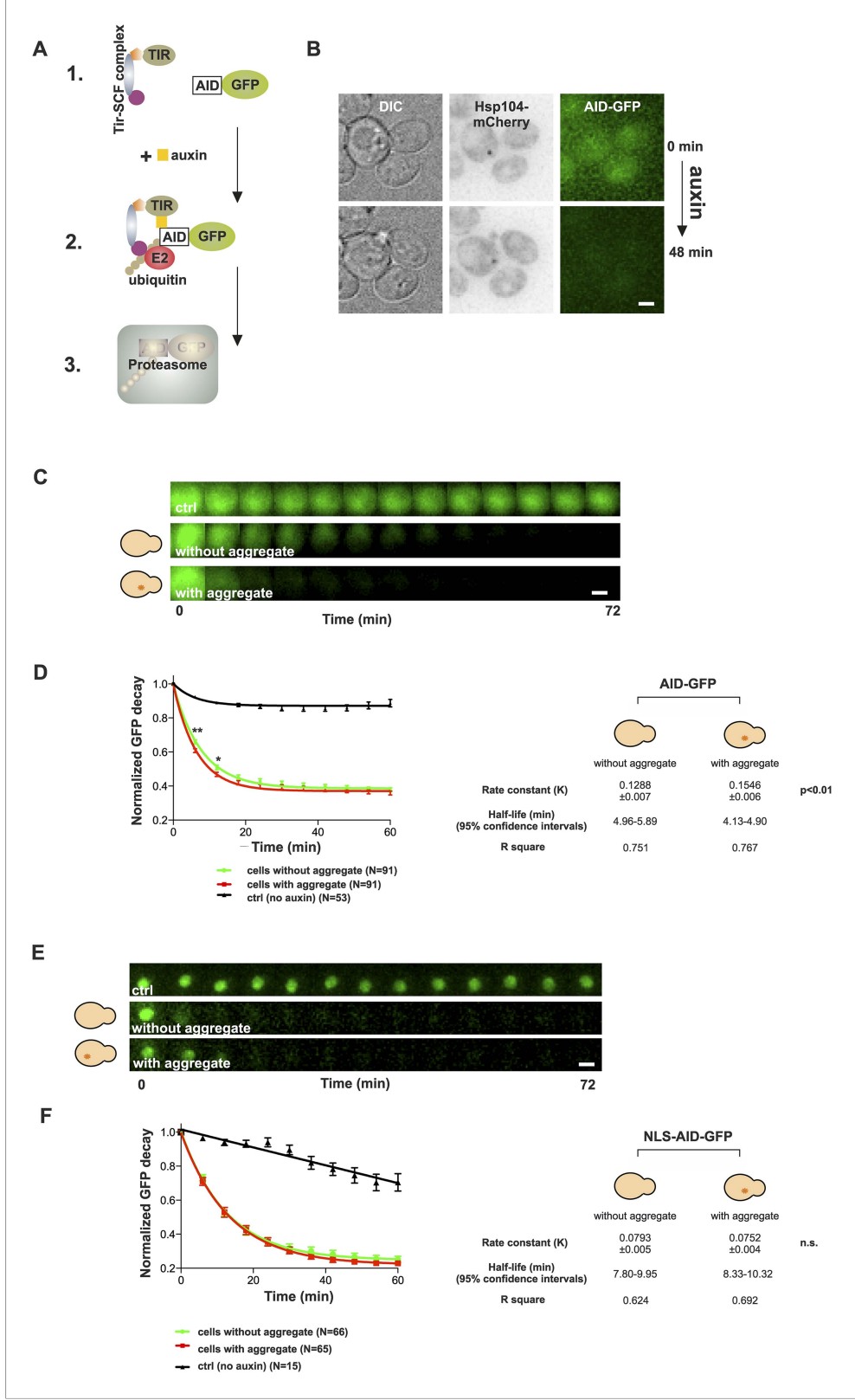

**Figure 5**. Presence of age-associated protein deposit promotes the function of cytosolic ubiquitin–proteasome system in vivo. (**A**) Schematic representation of the auxin-induced degron (AID)-system (*Nishimura et al., 2009*). Addition of auxin facilitates the recognition of the degron-motif in the target protein (GFP) by the exogenously expressed *Arabidobsis thaliana* E3 ligase SCF-Tir1 and subsequent ubiquitination and degradation by the
*Figure 5. continued on next page*

*Figure 5. Continued*

endogenous ubiquitin–proteaome system (UPS). (**B**) Representative images of cells expressing Hsp104-mCherry and AID-GFP taken immediately (0 min) and 48 min after auxin-addition. (**C**) Examples of time frames GFP degradation in the absence of auxin (upper panel) in the presence of auxin in cells with (middle) or without (below) an age-associated protein deposit. (**D**) The decay rates of GFP in the indicated groups. The error bars depict the normalized intensity values (average ± SEM) derived from 53 (ctrl) and 91 (auxin added) cells from three pooled replicates. The solid lines indicated non-linear one-phase decay fit. (**E**) GFP-NLS decay in the representative groups. (**F**) The decay rates and the graph fitting of NLS-GFP in the indicated groups as in (**D**). N = 15 (ctrl), 65–66 (auxin added) cells from three pooled replicates. The error bars depict the normalized intensity values (average ± SEM), while the solid lines indicate the non-linear one-phase decay fit.

its growth (*Figure 6D–F*). On the contrary, a significantly smaller fraction of cells over-expressing Hsp104 contained an age-associated protein deposit and while age-associated deposits still occasionally formed, they were rapidly destabilized (*Figure 6D–F*). Collectively, these data suggest that these chaperones functionally oppose each other (*Figure 6G*): the Hsp104/Hsp70 proteins prevent age-associated protein deposit formation possibly by refolding cytoplasmic substrates, whereas Hsp42 coordinates the collection of these substrates from the cytoplasm into the age-associated protein deposit (*Figure 6G*).

To consolidate this model, we further investigated the interplay between the refolding machinery (Hsp70/Hsp104) and the nucleation/growth promoting Hsp42. We used cells expressing Ssa1-GFP to visualize the age-associated protein deposit in the absence of Hsp104 (*Figure 7A*). Similarly to what we found for the deletion of *SSA1* and *SSA2* (*Figure 6B,C*), the *hsp104Δ* mutant cells formed the age-associated protein deposit more rapidly during aging. Again, the deletion of *HSP42* had an opposite effect, hampering formation of the Ssa1-GFP focus (*Figure 7A,B*). Remarkably, age-associated protein deposit formation was severely impaired in the *hsp104Δ hsp42Δ* double-mutant cells. These cells were crowded with punctate Ssa1-GFP (speckles), which failed to coalesce into a single regular age-associated protein deposit (*Figure 7A,B*). The portion of cells that contained multiple Ssa1-GFP foci was significantly higher in *hsp104Δ hsp42Δ* double-mutant cells than in *hsp104Δ* single-mutant cells. Interestingly, whereas >80% of wild-type and *hsp104Δ* mutant cells that contained an aggregate displayed a single Ssa1-GFP focus, the majority of *hsp42Δ* and *hsp104Δ hsp42Δ* mutant cells harbored several foci (*Figure 7C*).

To decisively dissect the individual contributions of Hsp104 and Hsp42 on age-associated protein deposit formation, we conceived an in vivo reconstitution assay where we could rapidly re-introduce either Hsp104 or Hsp42 to the aggregate-enriched *hsp104Δ hsp42Δ* double-mutant cells. To accomplish this, we mated *hsp104Δ hsp42Δ* double-mutant cells with wild-type (re-introducing both Hsp104 and Hsp42), *hsp104Δ* single-mutant (to reintroduce only Hsp42), *hsp42Δ* single-mutant (to reintroduce only Hsp104), or with the *hsp104 hsp42* double-mutant cells (reintroducing none of the chaperones) (*Figure 7D*). We monitored the deposit by imaging the fusing cells at 1-min intervals, using Ssa1-GFP as a reporter (*Figure 7D*). Importantly, simultaneous reintroduction of Hsp104 and Hsp42 was sufficient to clear the cytoplasm of Ssa1-GFP speckles already within 5 min (*Figure 7E*, no foci). In contrast, introducing Hsp42 alone resulted in rapid disappearance of cytoplasmic speckles with concurrent formation of a single Ssa1-GFP focus (*Figure 7E*, one focus). In contrary, reintroduction of only Hsp104 led to the slow and incomplete clearance of the Ssa-1 speckles, which was a reminiscent, but a milder phenotype as that observed upon conjugating the cell with *hsp104Δ hsp42Δ* double-mutant partner (*Figure 6E*, multiple foci). Collectively, this analysis supports the conclusion that Hsp42 functions to collect aggregates into the age-associated protein deposit structure, whereas Hsp104 functions to disaggregate/refold age-associated protein deposit destined substrates. In the context of the age-associated protein deposit cargo Sup35 (*Figure 2*), it is interesting to note that Hsp42 over-expression promotes [*PSI*+] curing, while its deletion results in enhanced [*PSI*+] induction, further supporting its role as the depositor for the age-associated protein deposit (*Duennwald et al., 2012*).

## Age-associated protein deposit formation is required for asymmetric inheritance of protein aggregates and may promote aging

We then asked whether age-associated protein deposit formation ensures the asymmetric inheritance of protein aggregates by the aging lineage. Examining its mitotic segregation (monitored with

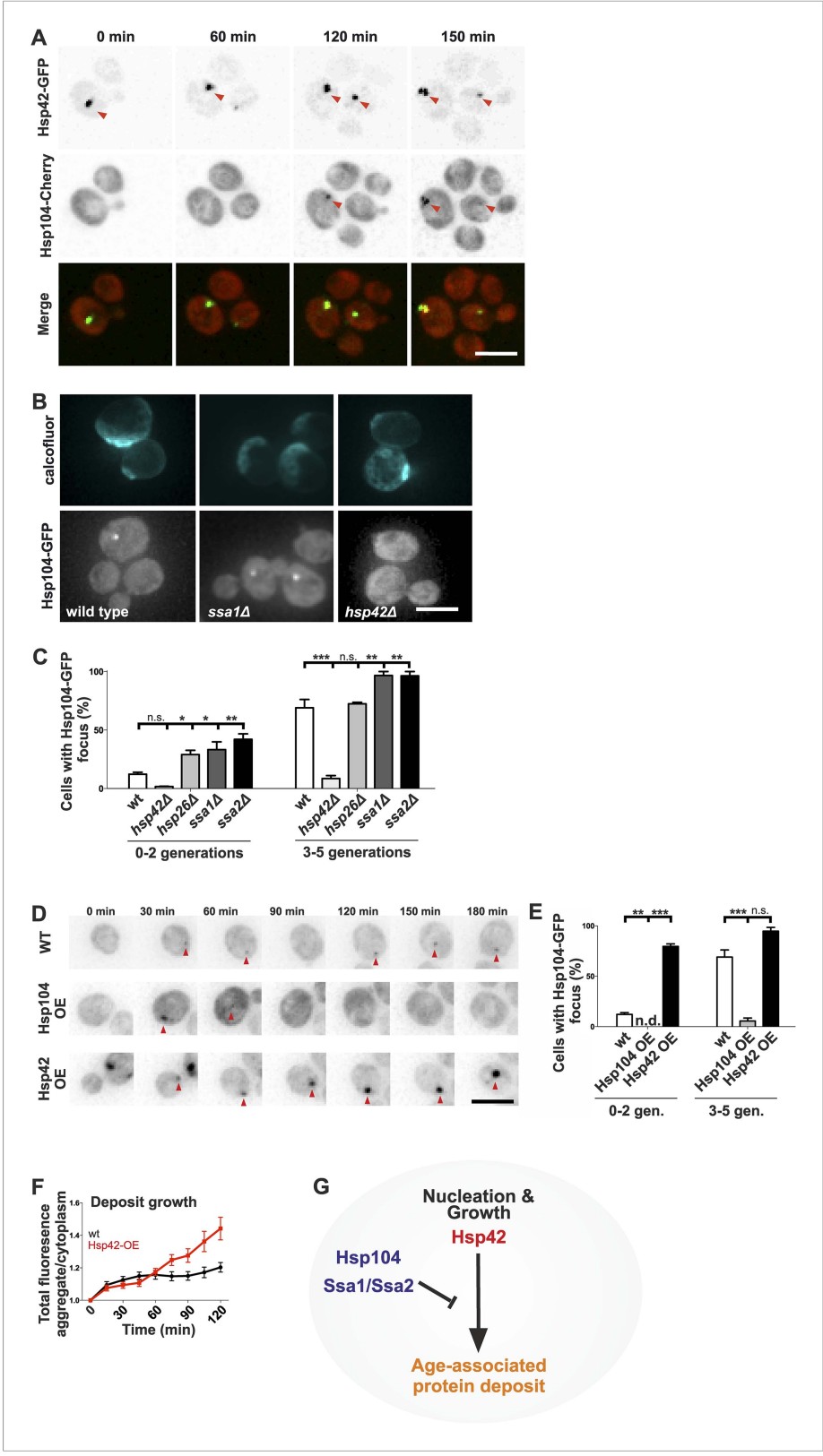

**Figure 6**. Identification of the roles of chaperones in age-associated protein deposit assembly. (**A**) Representative time-lapse images of cells expressing endogenously tagged Hsp42-GFP (upper panel) and Hsp104-mCherry (middle panel). Arrowhead depicts the age-associated protein deposit structure. (**B**) Representative images of aged Hsp104-GFP expressing wild-type, *ssa1Δ, and hsp42Δ* mutant cells. (**C**) Quantification of Hsp104-foci containing

*Figure 6. continued on next page*

*Figure 6. Continued*

cells in indicated age groups, (N = 317–1020 cells per genotype). (**D**) Time-lapse images of cells expressing endogenous Hsp104-GFP in wild-type, Hsp104 over-expressing, and Hsp42 over-expressing cells (age-associated protein deposit depicted by a red arrowhead). (**E**) Quantification of percentage of cells with age-associated protein deposit in wild-type, Hsp104 over-expressing, and Hsp42 over-expressing cells of indicated age groups (N = 432–1020 cells per genotype). (**F**) Quantification of endogenous Hsp104-GFP integrated density (age-associated protein deposit/cytoplasm) following its appearance in wild-type (black line) and Hsp42 over expressing cells (red line), (N = 19–23). (**G**) A summarizing model of the pathway underlying age-associated protein deposit formation. Scale bars 5 µm. Graphs display mean ± SEM, n.s not significant, *p < 0.05, **p < 0.01, ***p < 0.001.

Ssa1-GFP) in 96 wild-type and 186 *hsp104Δ* mutant cell divisions demonstrated that the age-associated protein deposit was in both cases inherited by the aging mother cell with high fidelity (in 98% of divisions; data not shown). In strong contrast, Ssa1-GFP foci were frequently inherited by the daughter cells when both *HSP104* and *HSP42* were deleted (*Figure 8A*, *Video 2*). By examining 61 mitotic events by time-lapse microscopy in the double-deleted cells, we observed Ssa1-GFP-foci relocating from the mother to the bud in 53 cases, but did not find any biased retrograde movement from the bud to the mother (*Fig. 8A*, *Video 2* and data not shown). Quantification of the percentage of buds with protein deposits demonstrated that more than 65% of all *hsp104Δ hsp42Δ* double-mutant buds displayed at least one aggregate, compared to less than 4% in *hsp104Δ* and *hsp42Δ* single-mutant and in wild-type cells (*Figure 8B*). Altogether, these data provide evidence that collection of protein aggregates into a single deposit by Hsp42 promotes their asymmetric retention in the mother cell during cell division.

Finally, we tested the significance of the age-associated protein deposit pathway and asymmetric protein aggregate segregation on replicative aging (*Figure 8C*). In accordance to previous work (*Erjavec et al., 2007*), mutant *hsp104Δ* cells were short-lived compared to wild-type cells (21.5 vs 27 generations, p < 0.001), supporting the idea that accelerated aggregate accumulation promotes aging. Further supporting this notion, deletion of *HSP42* extended the lifespan of the cells by more than 40% (39 generations, p < 0.001). The lifespan of the *hsp104Δ hsp42Δ* double-mutant cells was similar to that of the *hsp104Δ* single-mutant cells (23 vs 21.5 generations, respectively, ns.) and significantly shorter than that of the wild-type (p < 0.01) and *hsp42Δ* mutant cells (p < 0.001). The over-expression of *HSP42* had no effect on the lifespan (data not shown), suggesting that the wild-type levels of Hsp42 are sufficient to provide its maximal effects on life span, which is consistent with the appearance of the deposit early in life span in the wild-type cells. Together, these data imply that the Hsp104/Hsp70 system, which counteracts protein aggregation, is essential for longevity, while the Hsp42-dependent circuit promotes aging in the mother lineage, presumably by building the age-associated protein deposit and thereby establishing the asymmetric inheritance of age-associated protein aggregates.

## Discussion

Protein aggregation has been frequently associated with aging and aging-associated diseases. Here, by using the budding yeast as a model for replicative aging, we identify an Hsp42-promoted and Hsp104/Hsp70-counteracted pathway that deposits age-associated protein aggregates and thereby ensures their biased segregation to the aging mother-lineage upon cell division (see model in *Figure 7D*). This might represent a common mechanism to drive asymmetric segregation of aberrant proteins in dividing cells, as stressed fission yeast cells utilize a similar Hsp16-dependent mechanism for asymmetric partitioning of misfolded proteins (*Coelho et al., 2014*). Intriguingly, our data suggest that depositing age-associated protein aggregates to the mother cells might be a mixed blessing. On one hand, the age-associated deposit can promote spatial protein quality control (*Figures 4, 5*) and promote cell diversity as an asymmetrically dividing structure that may harbor protein conformation encoded epigenetic information, such as the prion fold of Sup35 (*Figure 2*). At the same time, the long-term consequences of this pathway appear negative for the aging lineage (*Figure 8C*). Why the deposit-forming pathway ultimately becomes associated with age-dependent loss of fitness remains to be determined. In this regard, it will be of great importance to elucidate how age-associated deposit relates to other aging pathways, such as those influencing the function of mitochondria

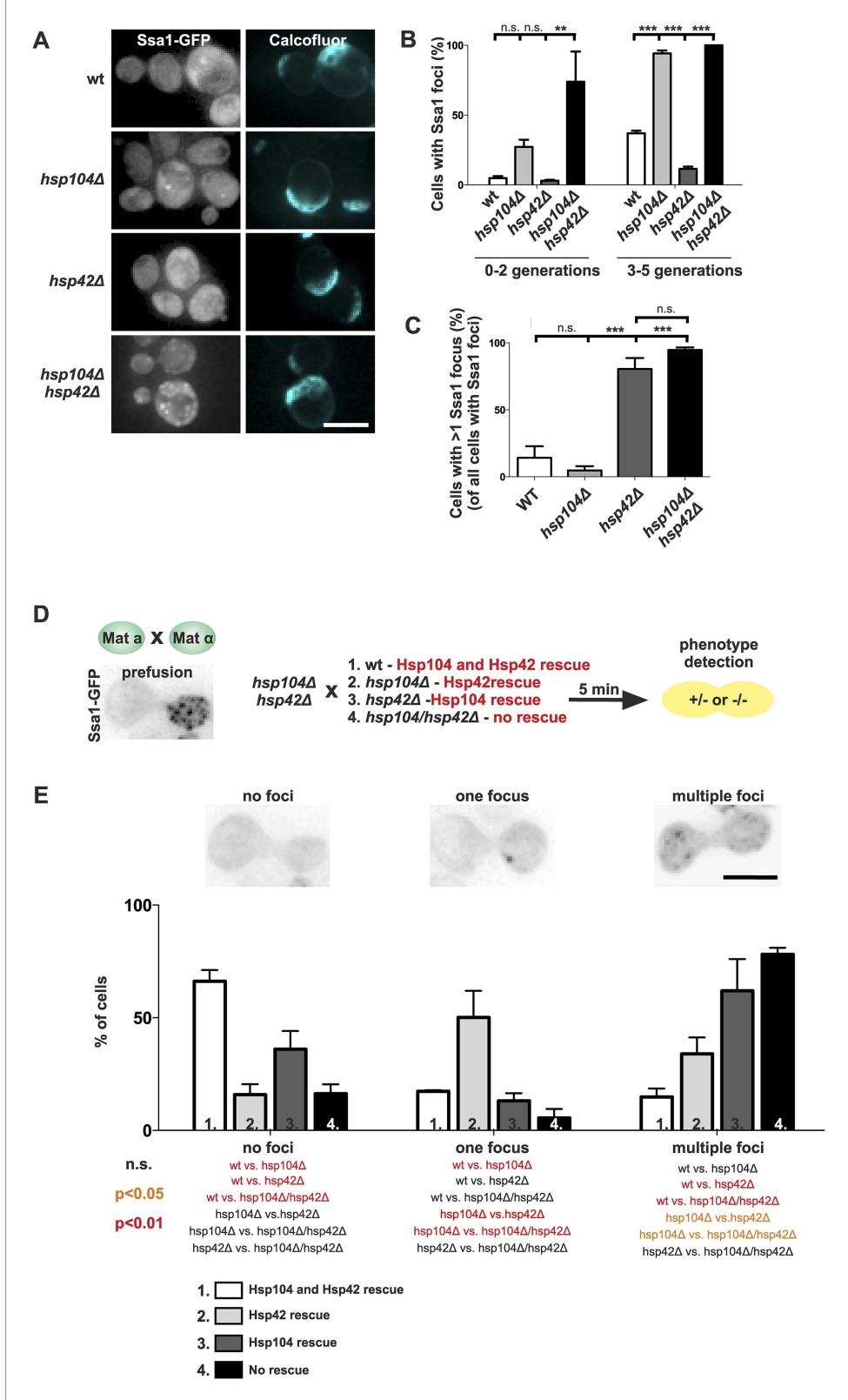

**Figure 7**. The assembly of age-associated protein deposit is promoted by Hsp42 and counteracted by Hsp104/
Hsp70. (**A**) Representative aged wild-type, *hsp104Δ*, *hsp42Δ* and *hsp104Δ* and *hsp42Δ* double-mutant cells.
(**B**) Quantification of the fraction of cells in indicated age groups that display Ssa1-GFP foci (age-associated protein
deposit), (N = 363–493 cells per genotype). (**C**) Quantification of the portion of cells that display >1 Ssa1 foci from all
*Figure 7. continued on next page*

*Figure 7. Continued*

Ssa1-foci containing cells, (N = 363–493 cells per genotype). (**D**) Illustration of the experimental scheme of the reconstitution assay. Cells that lack both *HSP104* and *HSP42* and display fragmented aggregate phenotype were mated with cells of the opposite mating type to reintroduce either Hsp104 and/or Hsp42. The resulting zygote was imaged with time-lapse microscopy and analyzed 5 min after fusion to score for aggregation phenotype (analyzed by Ssa1-GFP). (**E**) Quantification of the Ssa1-GFP phenotype (no foci, one focus, multiple foci) 5 min post fusion, (N = 73–100 fusion events per genotype). Scale bars 5 µm. Graphs display mean ± SEM, n.s not significant, **p < 0.01, ***p < 0.001.

(*McMurray and Gottschling, 2003*; *McFaline-Figueroa et al., 2011*), vacuoles (*Hughes and Gottschling, 2012*) and nuclear pores (*Shcheprova et al., 2008*; *Denoth-Lippuner et al., 2014*; *Webster et al., 2014*), and how the deposit localizes relative to these organelles and their components (*Chong et al., 2015*).

Protein deposit-containing cells efficiently coped with acute proteotoxic stress (*Figure 4*) and displayed improved degradation of cytosolic UPS substrate during early- to mid-life span (*Figure 5*). It is important to note, however, that our assays were conducted with relatively young cells that had recently acquired the protein deposit, and hence it is possible that UPS function starts to decline later during the aging process, as shown previously (*Andersson et al., 2013*). Curiously, the decline in proteostasis has been associated with protein aggregation pathologies such as Alzheimer's, Huntington's, and Parkinson's disease (*Vilchez et al., 2014*). The decreased degradation of cytosolic UPS substrates in a polyQ disease model was found to be due to the sequestration of the Hsp40 protein Sis1 (*Park et al., 2013*)—a protein that was not associated with the age-associated deposit (*Figure 1C*). It is thus plausible that the presence of the deposit might, for example, lead to increased levels of available (substrate unbound) Sis1, thereby allowing more efficient nuclear import and degradation of cytosolic UPS substrates. Altogether, these data suggest that there is a need to better differentiate between pathophysiological protein aggregation and 'homeostatic' protein aggregation that takes place during physiological aging, of which the latter might initially help to coordinate chaperones and UPS factors involved in protein quality control. Related to this, the age-associated protein deposit did not clearly fill out the set criterion for any of the previously characterized cytosolic deposits, including Q-bodies, JUNQ/INQ, or IPOD, but rather seems to exist in parallel with the JUNQ and IPOD structures (*Figure 1*, *Figure 3*)—although, for example, the chaperones regulating these deposits were often shared between these structures. This favors the notion that pathological aggregation processes (mimicked e.g., by VHL and Rnq1 over-expression) can hijack endogenous proteostasis regulatory mechanisms, possibly underlying their harmful effects in cells (*Park et al., 2013*). Altogether, these apparent differences between the stress-induced, pathological, and aging-associated protein aggregates (*Figure 2*, *Figure 3*) demonstrates that protein aggregates are not all equal in their composition or in the way they are being recognized and dealt by the cell and reinforces the importance to discriminate between different aggregation models.

In the future, it will be important to identify the cargo deposited to the age-associated protein deposits. In this aspect, it is interesting to note that many of the long-lived asymmetrically retained proteins that accumulate to mother cells as they age (*Thayer et al., 2014*) were among those localizing to the age-associated protein deposit (including Hsp104, Ssa1, Ssa2, and Hsp26). One example of an age-associated deposit resident protein was the prion-like Sup35. This storage rendered Sup35 to be inherited by the mother cells (unlike a prion, analogous to a mnemon (*Caudron and Barral, 2013*)) (*Figure 2*), which is consistent with the size-dependent transmission model of Sup35-aggregates (*Derdowski et al., 2010*). However, aging does not enhance Sup35-dependent [*PSI*+] prion generation (*Shewmaker and Wickner, 2006*). Together, these notions suggest that Sup35 cannot escape the age-associated deposit under normal growth conditions, perhaps owing to the rigid amyloid-like packing (*Saibil et al., 2012*), being consistent with the durable and non-dynamic nature of these deposits. However, this notion may come with a caveat: we found that the age-associated deposit could be, at least temporarily, disassembled by the heat shock response (*Figure 4*). This suggests the captivating possibility that changes in the environment might trigger the spread from the age-associated deposit. For cells facing fluctuating environments in the wild, the strategy to stochastically store prion-like proteins in aged cells might prove beneficial, as it could allow stress-dependent spread of the prion conformers from a subpopulation of aged-cells to their daughters, promoting faster adaptation to the changing environment (*Newby and Lindquist, 2013*).

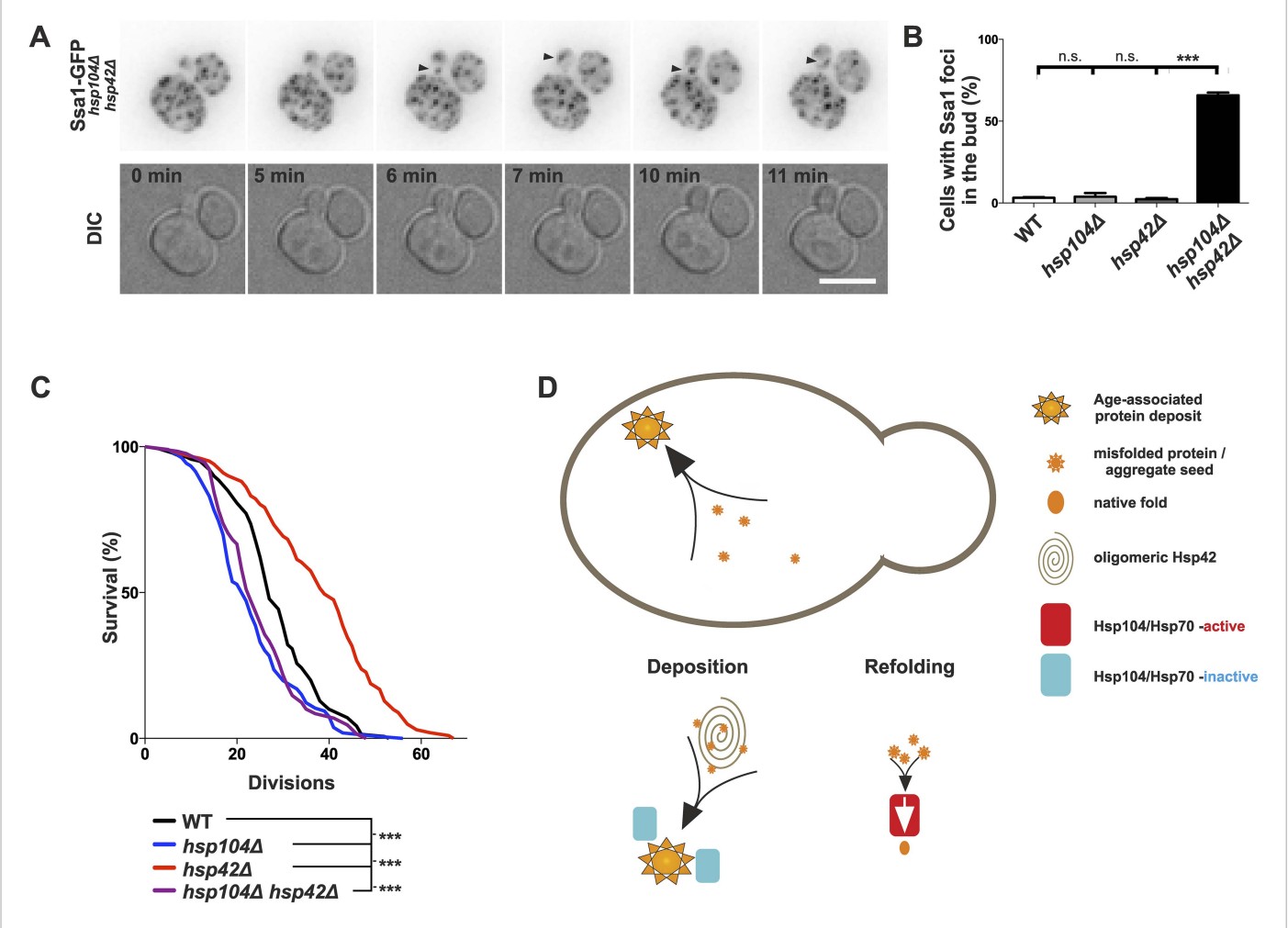

**Figure 8**. Age-associated protein deposit formation establishes asymmetric inheritance of protein aggregates and correlates with replicative age. (**A**) Time-lapse imaging of Ssa1-GFP in *hsp42Δ hsp104Δ* double-mutant cells (arrowheads denote Ssa1-GFP aggregates relocating from the mother cell to the bud). (**B**) Quantification of Ssa1-GFP foci found in the buds of mitotic yeast cells (N = 216–434 buds per genotype). (**C**) Replicative aging experiments of wild-type (black line, 27 generations), *hsp104Δ* mutant (blue line, 21.5 generations), *hsp42Δ* mutant (red line, 39 generations), and *hsp104Δ and hsp42Δ* double mutant (purple, 23 generations) single cells, (N = 101–140 cells per genotype). (**D**) A schematic model for the age-associated protein deposit pathway: Hsp42 acts as a collector of protein aggregate seeds and promotes their deposition at the ER membrane ensuring their asymmetric inheritance by the aging lineage during mitosis. These cytoplasmic seeds are subjected to Hsp104/Hsp70-dependent refolding, which is inactive (blue rectangle) at the site of the age-associated protein deposit assembly. Scale bar 5 µm. Graph displays mean ± SEM, n.s not significant, ***p < 0.001.

Since asymmetric inheritance of damaged proteins and protein aggregates seems to be conserved in metazoan stem cells (*Rujano et al., 2006*; *Bufalino et al., 2013*) that are also subjected to both replicative decline and segregation of lineages, we hypothesize that similar pathways that selectively segregate protein-aggregate based fate determinants asymmetrically during cell division are likely to be conserved and may contribute to cellular lineage specification across species.

## Materials and methods

### Strains and plasmids

Yeast strains used in this study are listed in *Supplementary file 1*. Strains were generated either manually (Janke et al., 2004) or derived from collections: (http://web.uni-frankfurt.de/fb15/mikro/euroscarf/col_index.html or http://clones.lifetechnologies.com/cloneinfo.php?clone=yeastgfp). Plasmids (pAG415-GDP) over-expressing Hsp104 (O-1360) and Hsp42 (O-2252), and the [*PIN*+][*PSI*+]

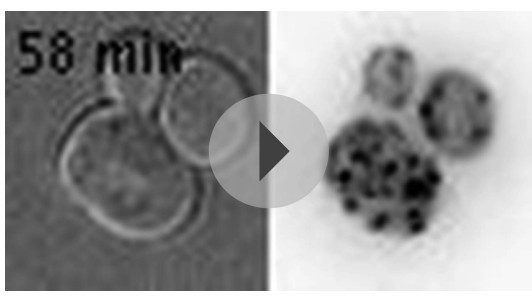

**Video 2.** The lack of the age-associated protein deposit assembly pathway results in the inheritance of the aggregates by the daughter cells. Expression of Ssa1-GFP was followed in *hsp104Δ hsp42Δ* double-mutant cell as it underwent cell division.

(yJW508) and [*PIN*+][*psi*–] (yJW584) parental strains (*Osherovich and Weissman, 2001*) were a generous gift from Simon Alberti (Max Planck Institute of Molecular Cell Biology and Genetics Dresden, Germany). The Mother Enrichment parental strains (*Lindstrom and Gottschling, 2009*) were a kind gift from Dan Gottschling (Fred Hutchinson Cancer Research Center, Seattle USA) and the plasmids encoding VHL-GFP and Rnq1-GFP were provided by Judith Frydman (via Addgene). The integrative plasmids for the auxin-mediated degradation assays (pADH1-OsTIR1-9myc, pADH1-eGFP-IAA17-NLS, pADH1-eGFP-IAA17) have been described in (Nishimura et al.) and were kindly provided by Matthias Peter (ETH Zurich, Switzerland).

## Cell cultures

Cells subjected to imaging were re-inoculated from overnight cultures to O.D. 600 nm 0.05 in YPD and grown to O.D. 0.5–0.8 at 30°C, centrifuged 500 g for 5 min, resuspended in synthetic complete (SC) media (-his), and mounted between a coverslip and an agar pad (SC-his). The over-expression of Rnq1-GFP and VHL-GFP was induced by switching cells to 2% galactose at the onset of imaging. The auxin-mediated degradation assays were performed similarly as above by placing cells to 0.5 mM 3-Indoleacetic acid (Sigma–Aldrich, #I2886) at the onset of imaging. The MEP was performed as described in (*Lindstrom and Gottschling, 2009*). Briefly, cells were grown in log phase for 15 hr before coupling the cell wall with Sulfo-NHS-LC-Biotin (Pierce). Following 2 hour recovery period, the MEP was engaged by addition of estradiol to a 1 µM final concentration. After desired incubation period, cells were coupled to uMACS Streptavidin MicroBeads (Miltenyi Biotec) and affinity purified with MACS separation columns (Miltenyi Biotec). Calcofluor staining was done by incubating cells with 5 µg/ml Fluorescent Brightener 28 (Sigma–Aldrich) for 1 min prior to centrifugation and resuspension to SC media. The Q-bodies were induced by incubating cells in a water bath at +42 °C for 30 min, or by using a temperature-controlled microfluidic chamber (see microfluidics). For the mating experiments, cells (1.85×10$^7$) of the opposite mating type were centrifuged 500 g for 5 min and resuspended to 40 µl SC media, mixed and immediately imaged with 1-min intervals (see microscopy for details).

## Microscopy

Wide-field microscopy was performed at 30 °C with a DeltaVision microscope coupled to a coolSNAP CCD HQ2 camera (Roper), 250W Xenon lamps and 100X/1.40 and 60x/1.42 NA Olympus oil immersion objectives. Z-sectioning was performed with 500-nm spacing (unless otherwise indicated), obtaining 9 or 11 (live-cell imaging) or 15 (calcofluor stained cells) stacks. Images were deconvolved with Softworx software (Applied Precision). Microfluidics on the aging chip were imaged with a DeltaVision microscope every 45 min for total of 66 hr by obtaining seven z-sections with 0.6-µm spacing. FRAP was done with temperature-controlled Zeiss LSM 510 microscope controlled with AIM LSM4.0 software, 63x 1.4NA Oil DIC Plan-Apochromat objective at 30°C, using DPSS and Argon lasers. Diploid cells expressing HSp104-GFP/Hsp104mCherry were imaged every 5 s at three z-planes with 800-nm spacing five times before bleaching the mCherry signal at the age-associated protein deposit with 100% DPSS laser power, after which the recovery of mCherry at the age-associated protein deposit was monitored over the period of two minutes by orienting with the GFP-signal.

## Microfuidics

The temperature ramp experiments (*Figure 4*) were carried out with ONIX microfluidic perfusion system equipped with a micro-incubator temperature controller CellASIC), using Y04C microfluidic plates (CellASIC). The temperature setup was the following: 30°C (50 min), 42°C (30 min) 30°C (300 min) and the cells were imaged every 10 min as described in (microscopy). The experiments were carried out in synthetic full medium with even flow rate of 2 psi. Microfluidics on the aging chip was performed with 1.5 µl/min flow rate as described in (*Denoth-Lippuner et al., 2014*).

## Image and statistical analyses

All image analyses were performed with Image J software (http://imagej.nih.gov/ij/). The aggregation foci were scored by eye from maximum intensity projected images (spanning the entire volume of the cell) and were defined as puncta that display high-intensity over the surrounding cytoplasmic background signal. For age-associated protein deposit intensity measurements, the integrated density was measured at the site over time at the age-associated protein deposit from stacked, non-processed sum-projected videos and was then normalized to the cytoplasmic Hsp104-GFP intensity.

For the auxin-mediated degradation experiments, the 11-plane z-stack covering 5.5 μm was sum projected and an integrated density of a defined area (18 μm$^2$ (GFP) or 16 μm$^2$ (GFP-NLS) was measured over time from the region of interest (ROI) and the neighboring background region (BG). To distinguish between cells that contained an aggregate from those that did not contain an aggregate, cells were categorized based on the first frame Hsp104-mCherry signal into the two respective groups. The ROI values were background subtracted [ROI(t)-BG(t)] and normalized to the first value [ROI(t$_x$)/ROI(t$_1$)]. The average of the pooled values was fitted with Prism5 software using non-linear one-phase decay.

For the FRAP analysis, raw data were background subtracted, corrected for acquisition-induced bleaching and normalized, and the curve was fitted from pooled values with Prism5 software using one-phase association non-linear fitting. The FRAP data where the aggregate structure was lost from the focal plane during recovery period image acquisition were discarded.

## Lifespan analysis

Lifespan of virgin daughters was analyzed on YPD plates using Zeiss Axioscope 40 microdissection microscope as previously described in (*Denoth-Lippuner et al., 2014*).

## Statistical analyses

All statistical analyses and graph preparations were done with Prism5 software. The error bars represent ±SEM from experimental triplicates with independent clones and statistical analyses were conducted with one-way ANOVA using Newman–Keuls post test, t-test, or Gehan-Breslow-Wilcoxon test.

## Acknowledgements

This study was financially supported by the ETH Zürich, FEBS Long-Term Postdoctoral Fellowship and a Finnish Cultural Foundation (Post Doc Pool) Fellowship (to JS) and through an advanced grant of the European Research Council to YB. We would like to thank Simon Alberti, Judith Frydman, Dan Gottschling, Björn Hegemann, and Matthias Peter for reagents, ScopeM, especially Tobias Schwarz for microscopic support, Marek Krzyzanowski, Sung-Sik Lee, and Bernhard Sebastian for kindly sharing their microfluidic expertize and Fabrice Caudron, Annina Denoth-Lippuner, and Ana-Maria Farcas for helpful comments on the manuscript.

## Additional information

### Funding

| Funder | Grant reference | Author |
|---|---|---|
| FEBS | Long Term Postdoctoral Fellowship | Juha Saarikangas |
| Suomen Kulttuurirahasto | Post Doc Pool | Juha Saarikangas |
| European Research Council (ERC) | BarrAge 250278 | Juha Saarikangas, Yves Barral |
| Eidgenössische Technische Hochschule Zürich | | Juha Saarikangas, Yves Barral |

The funders had no role in study design, data collection and interpretation, or the decision to submit the work for publication.

## Author contributions

JS, Conception and design, Acquisition of data, Analysis and interpretation of data, Drafting or revising the article; YB, Conception and design, Analysis and interpretation of data, Drafting or revising the article

## Additional files

**Supplementary file**

• Supplementary file 1. Yeast strains used in this study.

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
