## [Decision Letter]

[Editors’ note: this article was originally rejected after discussions between the reviewers, but the authors were invited to resubmit after an appeal against the decision.]

Thank you for choosing to send your work entitled “Early differentiation of the aging lineage in yeast by a protein deposit” for consideration at *eLife*. Your full submission has been evaluated by Randy Schekman (Senior Editor) and three peer reviewers, and the decision was reached after discussions between the reviewers. Based on our discussions and the individual reviews below, we regret to inform you that your work will not be considered further for publication in *eLife*.

We consider the work to be technically well done, but largely redundant with a lot of published literature – which you have not referenced properly. In addition there are specific concerns raised by the reviewers that we feel would require substantial experimental work, beyond the normal we suggest for a manuscript that could be returned with revisions. Consequently, in its current form, we believe this work does not advance the fields of aging or protein aggregation as far as other manuscripts that have been published recently in *eLife*. However, we do like the direction you have taken and think there are “good bones” in this work. Please take the advice of the reviewers into account in designing a new manuscript that you may wish to have us consider at a future date.

Reviewer #1:

The authors observe a single dot of Hsp104-GFP that forms in mother cells after several buddings, and is retained in the mother cell indefinitely. This focus is labeled as well by Ssa1 and Ssa2 (the major cytoplasmic Hsp70s), Hsp42 (a “small” heat shock protein) and Ydj1p (an Hsp40). These foci are shown to be distinct from “Q-bodies” (multiple aggregates formed after heat-stress) or “P-bodies”.

Critique:

1) There has been a proliferation of “foci” of protein aggregates. What is the relation of the AIDOPOD to the JUNQ (29) locus, both of which are juxtanuclear? Are these the same aggregates that Nyquist has described (without this name) as retained in the mother cell in an Hsp104-dependent process? Are these the Hsp42 foci of aggregated proteins described by Bukau's group (62)? Btn2p forms foci of prion (33) and non-prion aggregates (34) and binds well to Hsp42 (40): are the AIDOPOD foci the same as the Btn2 aggregation foci?

2) In the subsection “AIDOPOD formation is required for asymmetric inheritance of protein aggregates and may promote aging”, the authors state that “Altogether, these data provide evidence that collection of protein aggregates into a single deposit (AIDOPOD) by Hsp42 promotes their asymmetric retention in the mother cell during cell division.” This is not established by the data presented, which examines only the presence of various chaperones in these foci: Hsp104, Hsp70s (Ssa1p, Ssa2p), and Hsp42. Are there protein aggregates in these foci? Or do the authors mean that these are foci of aggregated Hsp104, Hsp70 and Hsp42? I doubt that. It is not established that there are any protein aggregates in the AIDOPOD foci, and they might even be membrane-bound collections of soluble proteins.

3) Figure 3: In [*PSI+*] cells, Sup35-GFP forms multiple dots (shown by many groups, but first by Patino and Lindquist in 1996). In Figure 3, there is just one big dot, and otherwise, the GFP seems to been even across the cell, just like the [*psi-*] cells in Figure 6. And Sup35-GFP dots in [*PSI+*] cells are seen in over 50% of the cells, not just in older cells. Something is wrong here.

4) Subheading “Formation of a protein deposit during early replicative aging”: “…Hsp104, a broad sensor for protein aggregates (Figure 1, (Jackrel et al, 2014)).” Jackrel et al do not show that a dot of Hsp104 means an aggregate of other proteins.

5) The Introduction is not very informative. How about Nyström's asymmetric protein segregation studies and aging? References are cited, but the results in the papers are not mentioned. How about Guarente's work on role of Sir2 and aging?

In summary, the work here is largely well done, but it seems likely that the authors' new name applies to foci described by Bukau, Nyström, Alberti, Kryndushkin, Kaganovich and others. And it is not clear that they are dealing with aggregates.

Reviewer #2:

This manuscript from Saarikangas and Barral characterizes age-induced protein aggregates that are asymmetrically retained in mother cells during replicative aging in *S. cerevisiae*. Their findings join a fairly extensive literature about protein aggregates, but the authors propose that the aggregates they define are distinct from previously described protein deposits and furthermore, that they contribute to replicative lifespan.

Background:

Protein aggregates have been observed and studied in cells for 50 years. But in the last nearly 20 years, there has been a steady appreciation that canonical heat-shock induced proteins are associated with aggregates (Lindquist lab and others). In yeast, Hsp104 was found to promote “disaggregation” of stress-induced protein aggregates, along with the aid of the Hsp70 family of chaperones and the Hsp26 and Hsp42 families of small heat-shock proteins.

More recently it has become clear that not all aggregates are equal. For instance, stress induced “Q bodies” initially form as numerous small protein aggregates that eventually coalesce into one or two larger, terminal aggregates (29; 17). Q-body formation is promoted by Hsp42 and antagonized by the disaggregase Hsp104. The best-characterized terminal protein aggregates are the juxtanuclear quality control (JUNQ) compartment and the vacuole-associated insoluble protein deposit (IPOD) compartment. JUNQ is composed of ubiquitinated misfolded proteins near proteasomes on the ER, whereas IPOD consists of vacuole-associated aggregates of terminally misfolded proteins that appear unable to be degraded.

Protein aggregates were first appreciated in yeast replicative aging when carbonylated proteins (a marker of oxidative damage – [1]) were found to be retained in the mother cell. These damaged proteins reside within Hsp104 foci (aggregates – [16]) and their retention in the mother is dependent upon Sir2 and actin.

More recently, the kinetics of Hsp104 foci formation after stress induction has been characterized (Zhou et al., 2014). The foci initiate during translation at the ER and ultimately form aggregates at the ER and mitochondria. The aggregates on mitochondria are retained in the mother in Fis1-dependent process.

Critique :

In the current manuscript, Saarikangas and Barral characterize an age-induced protein aggregate, which they call the AIDOPOD. It forms under normal, non-stressed conditions and is retained in mother cells. The experiments in this manuscript are technically sound and presented clearly. However, some of the data overlap substantially with previously published work, which is discussed below along with a list of the major findings.

1) AIDOPOD properties: forms naturally in old mother cells, usually within the first three generations; composed of heat-shock proteins Hsp104, Ssa1, Ssa2, Hsp42, and Ydj1, but not Hsp26, Sis1, or Hsp82; inherited asymmetrically to the mother.

Most of the AIDOPOD properties (age-induced foci, asymmetric retention) have been previously described for Hsp104-GFP foci. The current manuscript adds the observation that these foci co-localize with several other heat-shock proteins, although most of these markers have been characterized previously in stress-induced aggregates in young cells.

2) The AIDOPOD fusion-after-mating assays are largely novel and may represent a generally useful tool for studying the effects of different chaperones on protein aggregate formation/disassembly. However, as noted above, previous work has demonstrated by other means that stress-induced foci coalesce into larger aggregates, which presumably aids in asymmetric retention (17, 10).

3) AIDOPOD formation is Hsp42-dependent and antagonized by Hsp104.

A paradigm in which Hsp42 promotes aggregate formation while Hsp104 dissolves aggregates is already well-established and extensively tested in stress-induced aggregates (62, 17). The current manuscript extends this paradigm to age-induced protein aggregates.

4) *hsp42Δ* cells lacking an AIDOPOD are long-lived, whereas *hsp104Δ* cells, which form an AIDOPOD at an earlier age, are short-lived. *hsp104Δ* double mutants have many protein aggregate foci that distribute more symmetrically between daughter and mother, and these cells are as short-lived as the *hsp104Δ* single mutant.

The short lifespan of *hsp104Δ*mutants has been reported, as well as the inability of Hsp104 overexpression to extend lifespan (not reported here, but reported in [16]; [2]). The current authors appear to be the first to demonstrate and interpret the long lifespan of *hsp42Δ* mutants. The authors argue that Hsp104 is essential for a normal lifespan, whereas Hsp42, by aggregating misfolded or damaged proteins into the AIDOPOD in the mother, initiates mother/daughter aging asymmetry. Both aspects of this model have been proposed separately, but this work is the first to unify the models with regards to aging. A likely prediction from their model is that Hsp42 overexpression might shorten lifespan, as these cells accumulated an AIDOPOD at an earlier age than wild-type cells. The current manuscript would benefit substantially from this experiment.

Summary :

Taken together, the data presented here will provide a valuable resource for future studies on age-induced protein aggregates. However, much of the manuscript has merely extended existing models from protein aggregates that are stress-induced, particularly Q-bodies, to protein aggregates that are age-induced. Furthermore, previous studies on age-induced Hsp104 aggregates demonstrated that their asymmetric retention is actin and Sir2-dependent (16), and that their presence is intimately linked to proteasome function (2). The current manuscript ignores this existing framework with respect to characterizing the AIDOPOD and provides minimal mechanistic insight. Therefore, although valuable, the current manuscript does not substantially advance the field or lead to new paradigms, and would be better suited to a journal that accepts more modest advances.

Reviewer #3:

Yeast cells have been proposed to model diverse aspects of aging related to aging in humans. But this argument has failed to capture the imagination of colleagues studying other organisms. This paper could make an important difference.

Even more importantly, the asymmetric segregation of cell fate determinants is of great importance in biology and there are no systems to study this that are as tractable as yeast.

This paper provides an elegant demonstration of the organism's potential for rigorous thoughtful dissection of these problems. For example, the distinction between the age-related deposits and general stress deposits is surprising and important. The same is to be said for the distinction with P-bodies. The results of mating were fascinating. As were the results of the Hsp42 deletion and the interplay between Hsp42 deletion/OE and the deletion/OE of Hsp104, and of mating to further dissect their roles relative to that of Hsp70.

The paper is very well written (except see below) and the diversity and thoughtfulness of the experiments is very impressive.

That said, there are several points that need to be addressed.

Background information, writing, and referencing:

The paper has the abbreviated feeling of having been contributed to another journal with extreme pressure for publication space.

It should be re-written for *eLife*. For example, no information – or referencing – is provided about the biochemical function of Hsp42, which emerges as such a key player here. Nor is there any reference to phenotypes of Hsp42 mutants. In fact there are very few phenotypes reported, which makes the role of this protein in the aggregation foci particularly interesting.) There is virtually no referencing about Hsp104. Saibil's important work on Hsp104 and Ssa1 localization to specific unique foci in prion-containing cells isn't referenced. Nor are any of the foundational papers on Hsp104 function with Hsp70 in protein disaggregation.

Too many uses of “remarkably”.

Technical points:

Since so much of the work depends on the localization of fusion proteins, the ability of these proteins to rescue WT activity (that is, their functionality) should be referenced or established.

It was very convincing that Hsp104 “aggregation foci” remain in the mother cells. But it was very difficult to see the aggregation foci that are newly arising after several cell divisions in daughters that are becoming aging mothers – Figure 1, green arrowheads, and movies. This is such a crucial initial point, the evidence has to be convincing. The Materials and methods section here is very brief. Since these foci are difficult to see, was there an objective measure of the formation of these foci? For example, through quantitative high-content imaging? That is clearly implied by Figure 1, but not established. I have no doubt that the authors are seeing what they claim, but it isn't demonstrated adequately in the figures for readers to appreciate.

Biology:

A very substantial fraction of yeast cells contain a prion known as [*RNQ+*]. This prion is known to require Hsp104, to interact with Hsp104 and Hsp40, and to localize to specific cellular sub-compartments. Moreover, it is the one prion that most profoundly influences the aggregation of other proteins. It must be determined whether the cells the authors are studying contain this prion or not. And for at least the beginning phenomena, it should be looked at in [*RNQ+*] and [*rnq-*] cells.

I assume the cells they are analyzing for Sup35 foci are negative for the prion it forms? This should be stated. (It would be great to know if this mechanism of Sup35 concentration influenced prion formation. What is the effect of an Hsp42 KO? BUT this is outside the scope of this paper. I am just suggesting it to the authors as an interesting subject. I didn't find it at all surprising that Sup335 didn't influence Hsp104 organization – it shouldn't if the protein aggregating foci are a result of general aging. Perhaps the authors should explain the reasoning behind these observations.)

Is it accurate to compare the foci to an “organelle”? Or to call it a “compartment”? I think their own experiments suggest they are in much more dynamic association with cytoplasmic aggregates than is suggested by that terminology. I am asking the authors to think about whether this phrasing is necessary.

[Editors’ note: what now follows is the decision letter after the authors submitted for further consideration.]

Thank you for resubmitting your work entitled “Protein aggregates are associated with replicative aging without compromising protein quality control” for further consideration at *eLife*. Your revised article has been favorably evaluated by Randy Schekman (Senior Editor) and three reviewers. The manuscript has been improved but there are some remaining issues that need to be addressed before acceptance, as outlined below:

Reviewer #1 (Major Comments):

I believe that the authors have largely responded to my comments. The comparison of the current work's age-related aggregate site to other aggregate-collecting sites described by others is an important addition to this paper.

Reviewer #1 (Minor Comments):

The work of Nyström is referenced more extensively, but I am not sure there is a frank statement in the paper like that in the response to reviews: “It is likely that the Hsp104-labeled aggregates observed by [16] in aged cells refer to the same structure as those described in our work.”

Moreover, the statement at the end of the Introduction – “Our findings unveil a prevalent and highly coordinated early aging-associated aggregation response that opposes the view of aging and age-associated protein aggregation as a purely stochastic deterioration event” – I never thought it was thought to be “purely stochastic”. Rather, Nyström showed quite clearly that the retention of aggregates in the mother cell was very specific and directed. An opinion to the contrary from another group seems to have been refuted. And “unveils” is a little too strong.

Reviewer #2:

The authors have satisfactorily addressed the concerns I had with the earlier version of this manuscript. While there are similarities between the characterization of Hsp104 foci as described by Nyström and this work, the new characterization of the AIDOPOD and its differences between all the previously described stress induced foci (IPOD, JUNQ, Q-bodies) has never been explored before. Hence I feel it provides a bit of clarity to an otherwise confusing field – especially with respect to the aging field. Note that in the revised manuscript, the authors do a much better job of explaining differences between aging and external stress derived aggregates. Hence, I believe the manuscript is now suitable for publication in *eLife*.

Reviewer #3:

The paper is much improved. Its importance is nicely summarized in the last paragraph of the Introduction. I would also add that the pace with which protein foci are being reported in human diseases of aging, in yeast models of these proteinopathies, in the prion field, for diverse RNP granules, purinosomes, liquid-to-solid phase transitions of intrinsically disordered proteins, etc. has been accelerating at an amazing rate. Hence there is all the more need to have some distinctions and order brought to the field, for these distinct highly organized age-related deposits.

Wading into the realm of this literature is formidable for the naive reader. So I would suggest that the author note and comment on two recent papers that came out since they submitted their work.

Drummond's group has reported a fascinating formation of profuse, functional protein “aggregates” by particular groups of proteins in response to stress. It should be just mentioned that the current age-related deposits in this manuscript are distinct from those. (Also, see paragraph three, Introduction, the Drummond aggregates don't depend on translation).

Brenda Andrew’s group has reported on the dynamic behavior of many GFP protein foci and overlaps or distinctions should be mentioned.

---

## [Author Response]

[Editors’ note: the author responses to the first round of peer review follow.]

Following the reviewers’ advice, we have been able to substantially improve our manuscript, and would therefore like to request for the possibility to resubmit our work to *eLife* to be reconsidered for publication.

In the original submission, we described a physiological, early replicative aging-associated event in budding yeast: the formation of a singular and durable protein deposit harboring amyloid-like proteins, which is faithfully partitioned to the mother cell during cell division. We showed that the dynamics of this structure are dependent of the interplay between Hsp42-aggregase and Hsp104/Hsp70-disaggregase-activities. Furthermore, we demonstrated that the prevention of the deposit formation resulted in the inheritance of its constituents by the daughter cells, and prolonged the life span of the aging mother-cell-lineage, altogether providing evidence that protein aggregates are involved in the specification of the yeast-aging lineage.

All three reviewers of the original submission were overall positive about the quality of our work, but raised some concerns, which we have now addressed. In particular we have:

1. Expanded our analysis on the role of the age-associated protein deposit in protein quality control. We employed transient proteotoxic stress assays to show that aged cells with a protein deposit efficiently clear proteotoxic stress, and that exposure to proteotoxic heat stress counteracts the formation of the age-associated protein deposit. This provides evidence that the deposit formation is not a consequence of defective protein quality control.

2. Examined the relationship between the age-associated protein deposit and the previously described quality control “foci” (Q-bodies, JUNQ, IPOD and INQ). By visualizing these foci in the context of the age-associated deposit, we now provide evidence that the age-associated protein deposit does not fulfil the typical characteristics of any of the previously characterized protein deposit sites. This validates the originality of our findings and demonstrates an important distinction between age-induced, stress induced and pathological protein aggregation modes.

3. Analyzed the degradation kinetics of nuclear and cytoplasmic proteasome substrates at the single cell level in vivo. This demonstrated that the presence of the age-associated protein deposit aids the degradation of cytosolic proteasome substrates, providing evidence that this deposit has a functional role in promoting protein quality control. A point-by point-discussion is included below. Altogether, our results shed light on the poorly described aspects of aging, protein quality control, asymmetric cell division and cellular fate determination. We hope these extensive revisions will allow the reconsideration of our manuscript, now entitled “Protein aggregates are associated with replicative aging without compromising protein quality control”, for publication in *eLife*.

Reviewer #1:

Critique:

*1) There has been a proliferation of “foci” of protein aggregates. What is the relation of the AIDOPOD to the JUNQ (*[29]*) locus, both of which are juxtanuclear? Are these the same aggregates that Nyquist has described (without this name) as retained in the mother cell in an Hsp104-dependent process? Are these the Hsp42 foci of aggregated proteins described by Bukau's group (*[62]*)? Btn2p forms foci of prion (*[33]*) and non-prion aggregates (*[34]*) and binds well to Hsp42 (*[40]*): are the AIDOPOD foci the same as the Btn2 aggregation foci?*

As suggested by the reviewer, we have now addressed the relationship of the age-associated protein deposit with the previously described “foci”. Below, please find a detailed explanation of our results:

Relation to JUNQ/INQ:

We expressed the classical JUNQ-marker VHL-GFP and found that the resulting VHL-foci typically did not (87% of cases, N=29) recruit the age-associated deposit marker Hsp104 within 30 minutes from their appearance. In our hands, the VHL expression often led to an overall increase in Hsp104 expression, which made the overall interpretation regarding the age-associated deposit difficult (Figure 3).

As an alternative approach, we used a strain carrying Hsp82-GFP, Hsp104-mCherry and a deletion of rpn4, which results in a decreased amount of proteasomes and consequently in an accumulation of proteasome-destined material (JUNQ cargo). Hsp82 does not localize to the age-associated protein deposit (Figure 1), but localizes to Q-bodies (Figure 3), which are considered to be the early representation of JUNQ-destined material (17). Hsp104 marks Q-bodies (Figure 3), JUNQ (29) and the age-associated deposit (Figure 1). While most of the cells displayed several Q-body like aggregates enriched both with Hsp82 and Hsp104, we were also able to identify cells that displayed two discrete deposits: a JUNQ-like focus (Hsp82 and Hsp104 positive), and an age-associated deposit-like puncta (Hsp82 negative, Hsp104 positive) (see an example in Figure 3), suggesting that these two compartments can co-exist.

Relation to Btn2-foci (JUNQ/INQ):

The “JUNQ” was renamed as the “INQ” by a recent work showing that this compartment resides inside the nucleus and requires Btn2 for its formation (44).

We conducted rigorous analysis on the coexistence of Btn2 and Hsp104-deposits during aging. By analyzing altogether 1268 cells we found 30 examples where cells displayed a Btn2 focus, but only one of them was enriched in HSp104 (Figure 1, Figure 3). Furthermore, the appearance of the Btn2 focus was not associated with replicative aging (average age of Btn2-focus containing cells was 1.2 generations).

Relation to Q-bodies and the Hsp42 foci of aggregated proteins described by Bukau's group:

The study by [62] identified the requirement of Hsp42 for the formation of “peripheral aggregates”, which are analogous to the Q-bodies (17, 44). Our data shows that the Q-bodies and age-associated deposits can be distinguished by their durability (Figure 1 and Figure 3), and by the difference in associated chaperones (Figure 1 and Figure 3). Hence, although both structures require Hsp42 for their formation, they are likely not the same structures.

The JUNQ formation results from prolonged proteotoxic stress (formation of Q-bodies) associated with defective degradation (29, 17). We found that exposure to proteotoxic heat stress does not promote, but rather mildly inhibits the formations of age-associated protein deposits (Figure 4), suggesting that presence of JUNQ destined material does not promote age-associated deposit formation.

Relation to IPOD:

We also expanded this analysis to look at the IPOD structure (undynamic aggregate formed by Rnq1, [29]). In contrast to what we observed with the JUNQ marker VHL, the Rnq1-aggregates were efficiently recognized by Hsp104 (>98% of Rnq1-foci accumulated Hsp104 within 30 min of their appearance, N=53). However, we did not observe accumulation of Rnq1 to a pre-existing age-associated protein deposit (see arrowhead in Figure 3). Moreover, we analyzed 117 Hsp104 foci-containing [*PIN+*, *PSI+*]-cells and found that in 98.3% of the cases, Rnq1-GFP did not accumulate to this site (Figure 3). The two examples of cells that had and Rnq1-focus, both displayed >1 Hsp104 foci, of which only one was Rnq1-positive (Figure 3). Together with the results of the Rnq1-over-expression, this suggests that the Rnq1-aggregate is probably not equivalent to the “typical” age-associated deposit.

Relation to the age-associated aggregates that Nyström lab has described:

It is likely that the Hsp104-labeled aggregates observed by [16] in aged cells refer to the same structure as those described in our work. There are some differences regarding the appearance (the number of foci per cell appears higher in their work) and segregation (we did not observe a segregation defect of age-associated aggregates in hsp104 mutant cells). These disparities could be due to different experimental approaches used by these studies.

Although we acknowledged their seminal papers in the original submission, we have now paid attention to discuss them in more detail in the Introduction and refer to them where appropriate.

Collectively, these results provide evidence that the age-associated deposit is not identical with Q-bodies, JUNQ/INQ, Btn2-foci, or IPOD, but rather that it can co-exist with these deposit sites. These results are now shown in Figure 1, Figure 3, Figure 4 and discussed in the Results and Discussion.

2) In the subsection “AIDOPOD formation is required for asymmetric inheritance of protein aggregates and may promote aging”, the authors state that “Altogether, these data provide evidence that collection of protein aggregates into a single deposit (AIDOPOD) by Hsp42 promotes their asymmetric retention in the mother cell during cell division.” This is not established by the data presented, which examines only the presence of various chaperones in these foci: Hsp104, Hsp70s (Ssa1p, Ssa2p), and Hsp42. Are there protein aggregates in these foci? Or do the authors mean that these are foci of aggregated Hsp104, Hsp70 and Hsp42? I doubt that. It is not established that there are any protein aggregates in the AIDOPOD foci, and they might even be membrane-bound collections of soluble proteins.

We show that the age-associated deposit collects endogenous protein aggregates formed by Sup35 in its prion [*PSI+*] form, and that this Sup35-enriched deposit is asymmetrically retained by the mother cells during cell division (Figure 2). Since the non-prion [*psi-*] form of Sup35-GFP did not accumulate to the age-associated deposit, we assume that the deposit recruits amyloid-like substrates. These results are now described in the revised manuscript in Figure 2 and we have further clarified this issue in the text.

*3)*
Figure 3*: In [*PSI+*] cells, Sup35-GFP forms multiple dots (shown by many groups, but first by*
[53]*. In*
Figure 3*, there is just one big dot, and otherwise, the GFP seems to been even across the cell, just like the [psi-] cells in*
Figure 6*. And Sup35-GFP dots in [*PSI+] *cells are seen in over 50% of the cells, not just in older cells. Something is wrong here.*

We have now clarified this issue by improving the quality of the original pictures to allow a more accurate interpretation of the phenotype of the [*PSI+*] cells. As pointed out by the reviewer, Sup35 often forms multiple aggregates in [*PSI+*] cells (Figure 2, white arrowhead indicates Sup35 in the age-associated protein deposit, red arrowhead indicates Sup35 aggregate seeds that are not associated with Hsp104). Most of the studies that have visualized Sup35-dots (including the study by Patino et al.) made use of over-expression constructs, which often encode solely the prion domain of Sup35 (Sup35-NM). This is, for many reasons, not comparable to visualization of the endogenous protein, as was done in our study.

Our findings are in agreement with studies conducted with endogenously tagged Sup35 in [*PSI+*], showing that the aggregate distribution depends on the strength of the [*PSI*]-phenotype (“strong” [*PSI+*] has more small aggregates and “weak” [*PSI+*] fewer but larger aggregates) and, importantly, that replicative aging result in shifting towards larger Sup35 aggregates (see e.g. Derdowski et al., 2011).

*4) Subheading “Formation of a protein deposit during early replicative aging”: “…Hsp104, a broad sensor for protein aggregates (*Figure 1*, (Jackrel et al, 2014)).” Jackrel et al do not show that a dot of Hsp104 means an aggregate of other proteins.*

We have now changed the reference for this statement to a recent review ([20].). This review summarizes the role and numerous substrates of Hsp104-mediated protein disaggregation.

5) The Introduction is not very informative. How about Nyström's asymmetric protein segregation studies and aging? References are cited, but the results in the papers are not mentioned. How about Guarente's work on role of Sir2 and aging?

As discussed above, we have now re-written and substantially expanded the Introduction to make sure we discuss and reference the results of all the publications that are relevant to our work.

Reviewer #2:

Critique:

1) AIDOPOD properties: forms naturally in old mother cells, usually within the first three generations; composed of heat-shock proteins Hsp104, Ssa1, Ssa2, Hsp42, and Ydj1, but not Hsp26, Sis1, or Hsp82; inherited asymmetrically to the mother.

Most of the AIDOPOD properties (age-induced foci, asymmetric retention have been previously described for Hsp104-GFP foci. The current manuscript adds the observation that these foci co-localize with several other heat-shock proteins, although most of these markers have been characterized previously in stress-induced aggregates in young cells.

We fully agree with the reviewer that it has been established that protein aggregates accumulate in aging cells, and made sure to further emphasize this in the Introduction and Results. However, we do not share the notion that most of the properties we characterized for the age-associated deposit would have been characterized before. We appreciate the reviewer for these sentiments, as it helped us to recognize that we were not able to phrase our findings and conclusions clearly and gave us an opportunity to clarify these aspects. We have now revised on our paper substantially and focused on what we feel are the key unresolved issues in the field. In particular, owing to the variable use of reporters and model systems, it has remained unclear to which extend the age-associated protein aggregation can be modeled by these reporter systems (i.e. whether they truly represent the same pathway or rather parallel pathways that share the same molecular components). Thus we have endeavored to answer the following questions:

1. What is the relationship between the age-associated protein aggregation and the formation of Q-bodies, IPOD and JUNQ/INQ?

2. What is the role of the age-associated protein deposit in the functionality of protein quality control?

These efforts resulted in the following results, which are now included in the work:

The age-associated deposit is not analogous the Q-bodies, JUNQ/INQ or IPOD (Figure 3, see response to reviewer #1).

Cells with an age-associated deposit respond and dissolve proteotoxic stress as fast as their clonal counterparts without an aggregate, providing the evidence that their formation is not due to inability of cells to clear protein aggregates (Figure 4).

Proteotoxic stress (Q-bodies) does not enhance, but rather counteracts the formation of the age-associated deposit (Figure 4).

Cells with an age-associated protein deposit display accelerated degradation rate of cytosolic proteasome substrate, demonstrating a direct functional role for the age-associated protein aggregation (Figure 5).

Together, these added data provide evidence that age-associated protein aggregation has initially a positive influence on protein quality control and resolves its relation to the previously described deposit sites.

*2) The AIDOPOD fusion-after-mating assays are largely novel and may represent a generally useful tool for studying the effects of different chaperones on protein aggregate formation/disassembly. However, as noted above, previous work has demonstrated by other means that stress-induced foci coalesce into larger aggregates, which presumably aids in asymmetric retention (*[17]*,*
[10]*).*

Please see our reply below.

3) AIDOPOD formation is Hsp42-dependent and antagonized by Hsp104.

*A paradigm in which Hsp42 promotes aggregate formation while Hsp104 dissolves aggregates is already well-established and extensively tested in stress-induced aggregates (*[62]*,*
[17]*). The current manuscript extends this paradigm to age-induced protein aggregates.*

Indeed, our data supports the proposed aggregase-disaggregase-scheme, which seems generally apply to stress-induced aggregation in both budding and fission yeasts (62, 17, 10). Our work brings upon one important distinction to this model: unlike with stress associated aggregates, in the context of aging, Hsp104 is unable to dissolve the age-associated structures (Figure 1) – unless cells experience stress response, which can lead to transient disassembly of the age-associated deposit (Figure 4). We have discussed this in the model presented in Figure 8, as well as in the Results and Discussion.

*4)* hsp42Δ *cells lacking an AIDOPOD are long-lived, whereas cells, which form an AIDOPOD at an earlier age, are short-lived.* hsp104Δ *double mutants have many protein aggregate foci that distribute more symmetrically between daughter and mother, and these cells are as short-lived as the* hsp104Δ *single mutant.*

*The short lifespan of* hsp104Δ *mutants has been reported, as well as the inability of Hsp104 overexpression to extend lifespan (not reported here, but reported in*
[16]*;*
[2]*). The current authors appear to be the first to demonstrate and interpret the long lifespan of* hsp42Δ *mutants. The authors argue that Hsp104 is essential for a normal lifespan, whereas Hsp42, by aggregating misfolded or damaged proteins into the AIDOPOD in the mother, initiates mother/daughter aging asymmetry. Both aspects of this model have been proposed separately, but this work is the first to unify the models with regards to aging. A likely prediction from their model is that Hsp42 overexpression might shorten lifespan, as these cells accumulated an AIDOPOD at an earlier age than wild-type cells. The current manuscript would benefit substantially from this experiment.*

As the reviewer suggested, we have now measured the life span of Hsp42 over-expressing cells. This pedigree analysis showed that Hsp42 over-expression did not have any effect on median life span compared to wild type cells (22 vs. 22 divisions, N=141 each, see below). This suggests that the effect of Hsp42 on aging is already fully attained with the wild type protein levels, which is not surprising since most cells in both groups (WT and Hsp42-OE) formed the deposit already during the first quarter of their life span (Figure 6). In this regard, it is important to note that we still do not exactly understand how does the deposit influence life span. We have now discussed this result in the subsection “Age-associated protein deposit formation is required for asymmetric inheritance of protein aggregates and may promote aging”.

Author response image 1.**DOI:**
http://dx.doi.org/10.7554/eLife.06197.018

Summary:

*Taken together, the data presented here will provide a valuable resource for future studies on age-induced protein aggregates. However, much of the manuscript has merely extended existing models from protein aggregates that are stress-induced, particularly Q-bodies, to protein aggregates that are age-induced. Furthermore, previous studies on age-induced Hsp104 aggregates demonstrated that their asymmetric retention is actin and Sir2-dependent (*[16]*), and that their presence is intimately linked to proteasome function (*[2]*). The current manuscript ignores this existing framework with respect to characterizing the AIDOPOD and provides minimal mechanistic insight. Therefore, although valuable, the current manuscript does not substantially advance the field or lead to new paradigms, and would be better suited to a journal that accepts more modest advances.*

We have now extensively discussed the existing framework in the Introduction, Results and Discussion. We also provide experimental proof that clarifies the important contextual/functional differences between age-associated protein aggregation and stress/pathological protein aggregation models. Furthermore, we provide evidence supporting a functional role for age-associated protein aggregation in cellular quality control.

Reviewer #3:

Background information, writing, and referencing:

The paper has the abbreviated feeling of having been contributed to another journal with extreme pressure for publication space.

*It should be re-written for* eLife*. For example, no information – or referencing – is provided about the biochemical function of Hsp42, which emerges as such a key player here. Nor is there any reference to phenotypes of Hsp42 mutants. In fact there are very few phenotypes reported, which makes the role of this protein in the aggregation foci particularly interesting.) There is virtually no referencing about Hsp104. Saibil's important work on Hsp104 and Ssa1 localization to specific unique foci in prion-containing cells isn't referenced. Nor are any of the foundational papers on Hsp104 function with Hsp70 in protein disaggregation.*

Actually, this work has never been submitted elsewhere. We want to apologize for the tight formatting of our original submission, which was an oversight on our part due to an initial intent to submit this work as an *eLife* “short report”. However, prior to submission we realized that we would not be able to fulfill the length requirements of this format. We have now re-written the paper in the article format and made sure we discuss and reference the most important papers related to our work, including those kindly pointed out by the reviewer.

Too many uses of “remarkably”.

We have paid attention to reduce this wording.

Technical points:

Since so much of the work depends on the localization of fusion proteins, the ability of these proteins to rescue WT activity (that is, their functionality) should be referenced or established.

The main reference proteins used in this study are Hsp104 and Ssa1 tagged C-terminally with a fluorophore (GFP or mCherry). The Hsp104-C-terminal fluorophore chimera has previously been shown to be fully functional in disaggregation (thermotolerance) (see e.g. [62], [9]). We have now referenced to this in the Figure 1 legend. Regarding the functionality of Ssa1-GFP, we have crossed it to *ydj1Δ*, which did not lead to lethality, although according to the SGD (http://www.yeastgenome.org/), the ssa1 mutant is synthetic lethal with *ydj1Δ*. This argues that Ssa1-GFP is functional, at least in the conditions used in our study.

*It was very convincing that Hsp104 “aggregation foci” remain in the mother cells. But it was very difficult to see the aggregation foci that are newly arising after several cell divisions in daughters that are becoming aging mothers –*
Figure 1*, green arrowheads, and movies. This is such a crucial initial point, the evidence has to be convincing. The Materials and methods section here is very brief. Since these foci are difficult to see, was there an objective measure of the formation of these foci? For example, through quantitative high-content imaging? That is clearly implied by*
Figure 1*, but not established. I have no doubt that the authors are seeing what they claim, but it isn't demonstrated adequately in the figures for readers to appreciate.*

The aggregation foci were scored by eye from maximum intensity projected images (spanning the entire volume of the cell), and were defined as puncta that display high-intensity over the surrounding cytoplasmic background signal. The quantification of aggregate intensities were carried out from SUM projected videos, and the intensity values were normalized to the surrounding cytoplasmic value. The formation of the age-associated deposit was followed in time-lapse movies, where their dynamic maturation process could be followed frame-by-frame (Figure 1). We have now improved the quality of the images in Figure 1 and have expanded the Materials and methods to describe in more detail how the aggregates were detected.

Biology:

*A very substantial fraction of yeast cells contain a prion known as [*RNQ+*]. This prion is known to require Hsp104, to interact with Hsp104 and Hsp40, and to localize to specific cellular sub-compartments. Moreover, it is the one prion that most profoundly influences the aggregation of other proteins. It must be determined whether the cells the authors are studying contain this prion or not. And for at least the beginning phenomena, it should be looked at in [*RNQ+*] and [*rnq-*] cells.*

We have now looked at the influence of the [*rnq1-*] vs. [*RNQ+*]-state on the formation of the age-associated protein deposit.

To distinguish if the Mother Enrichment Program (MEP) strain mainly used in this study is in the [*rnq-*] or in the [*RNQ+]-*state, we expressed the Sup35-NM-domain in these cells. The Sup35-NM expression results in the formation of aggregates only if the cell is in the [*RNQ+*]-state (see e.g. Liebman et al., 2006, Methods). The MEP-strain readily formed aggregates upon induction of Sup35-NM-YFP-expression (see below) and hence is in the [*RNQ+*]-state. We have now mentioned this in the text (subheading “The physiological constituents of the age-associated protein deposit include prion protein Sup35”).

Author response image 2.The expression of Sup35-NM-YFP in the MEP strain.Left: at the start of the GAL induction, Right: 250 minutes later.**DOI:**
http://dx.doi.org/10.7554/eLife.06197.019

We also assessed the influence of the [*rnq-*] state on the age-associated protein aggregate formation, by comparing the Hsp104-deposit formation between rnq1Δ [*rnq- psi-*] and WT [*RNQ*+ *PSI+*] cells. We found that the rnq1Δ cells displayed an enhanced formation of age-associated Hsp104-foci as compared to the [*RNQ+*]-cells (N=118-874 cells per group, see below).

Author response image 3.**DOI:**
http://dx.doi.org/10.7554/eLife.06197.020

Finally, we also looked at the localization of Rnq1-GFP to the age-associated deposit and found that it typically does not localize there, although when it aggregates, it is recognized by Hsp104 (please see the detailed discussion above in the answer to Referee #1).

We find it very interesting and somewhat unexpected that the deletion of rnq1 [*psi-*] has a positive effect on aggregate appearance. However, since we do not yet fully understand why this is and how it relates to our existing framework, we have decided to not include these data into the current manuscript, but mentioned this result in subheading “The physiological constituents of the age-associated protein deposit include prion protein Sup35”.

I assume the cells they are analyzing for Sup35 foci are negative for the prion it forms? This should be stated. (It would be great to know if this mechanism of Sup35 concentration influenced prion formation. What is the effect of an Hsp42 KO? BUT this is outside the scope of this paper. I am just suggesting it to the authors as an interesting subject. I didn't find it at all surprising that Sup335 didn't influence Hsp104 organization – it shouldn't if the protein aggregating foci are a result of general aging. Perhaps the authors should explain the reasoning behind these observations.)

The cells that accumulate Sup35 into the age-associated foci are in the prion [*PSI+*] form. In the prion negative [*psi-*] cells, Sup35 did not accumulate to the age-associated deposit, suggesting that prion (amyloid)-like fold directs it to be stored at the age-associated protein deposit. This also fits with many characteristics of this deposit, including its durability and the slow exchange rate.

Interestingly, the deletion of Hsp42 promotes [*PSI+*] prion induction and its over expression promotes curing (15), which is in accordance with the concept that it functions as the depositor. We have now expanded the discussion about the possible outcome of Sup35 storage to the age-associated deposit (paragraph three, Discussion).

Is it accurate to compare the foci to an “organelle”? Or to call it a “compartment”? I think their own experiments suggest they are in much more dynamic association with cytoplasmic aggregates than is suggested by that terminology. I am asking the authors to think about whether this phrasing is necessary.

Following the advice of the reviewer, we have revised the text and no longer talk about the structure as “organelle-like”, but rather as a compartment.

[Editors’ note: the author responses to the re-review follow.]

Reviewer #1 (Minor Comments):

*The work of Nyström is referenced more extensively, but I am not sure there is a frank statement in the paper like that in the response to reviews: “It is likely that the Hsp104-labeled aggregates observed by*
[16]
*in aged cells refer to the same structure as those described in our work.”*

In the Results we write as follows:

“Interestingly, we found many cells displaying an aggregate (typically a single bright Hsp104-labelled focus) and this portion increased in a progressive, age-dependent manner such that >80% of cells that had undergone more than 6 divisions displayed such a structure (Figure 1), as previously reported ([1] and [16])”.

Moreover, the statement at the end of the Introduction – “Our findings unveil a prevalent and highly coordinated early aging-associated aggregation response that opposes the view of aging and age-associated protein aggregation as a purely stochastic deterioration event” – I never thought it was thought to be “purely stochastic”. Rather, Nyström showed quite clearly that the retention of aggregates in the mother cell was very specific and directed. An opinion to the contrary from another group seems to have been refuted. And “unveils” is a little too strong.

We have now modified the end of the Introduction according to the reviewer’s suggestion.

Reviewer #3:

Wading into the realm of this literature is formidable for the naive reader. So I would suggest that the author note and comment on two recent papers that came out since they submitted their work.

Drummond's group has reported a fascinating formation of profuse, functional protein “aggregates” by particular groups of proteins in response to stress. It should be just mentioned that the current age-related deposits in this manuscript are distinct from those. (Also, see paragraph three, Introduction, the Drummond aggregates don't depend on translation).

We have now cited this interesting paper by the Drummond group in (in the Introduction and subheading “The age-associated deposit is distinguishable from the previously described protein quality control deposits”), where we compare the heat-induced and age-associated aggregates. Furthermore, we decided to remove the sentence in the Introduction about the requirement of translation to the heat induced aggregates.

Brenda Andrews group has reported on the dynamic behavior of many GFP protein foci and overlaps or distinctions should be mentioned.

We are not certain to which paper the reviewer is referring to, but we suspect it to be the recent paper by the Andrews group concerning the proteome wide localization analysis (7). This database and its source files will surely be a valuable resource to search for proteins that might be directed to the age-associated protein deposit. However, their automated identification pipeline of protein localization did not include cellular puncta, as stated in this paper: “small punctate compartments are difficult to distinguish computationally”. We have now cited this paper in the Discussion, where we discuss the importance to compare the localization of the age-associated aggregate with organelles and their components.